# Endocytosis regulates TDP-43 toxicity and turnover

Guangbo Liu[1], Alyssa N. Coyne[1], Fen Pei[1], Spencer Vaughan [1], Matthew Chaung[1], Daniela C. Zarnescu[1,2] & J. Ross Buchan[1]

Amyotrophic lateral sclerosis (ALS) is a fatal motor neuron degenerative disease. ALS-affected motor neurons exhibit aberrant localization of a nuclear RNA binding protein, TDP-43, into cytoplasmic aggregates, which contributes to pathology via unclear mechanisms. Here, we demonstrate that TDP-43 turnover and toxicity depend in part upon the endocytosis pathway. TDP-43 inhibits endocytosis, and co-localizes strongly with endocytic proteins, including in ALS patient tissue. Impairing endocytosis increases TDP-43 toxicity, aggregation, and protein levels, whereas enhancing endocytosis reverses these phenotypes. Locomotor dysfunction in a TDP-43 ALS fly model is also exacerbated and suppressed by impairment and enhancement of endocytic function, respectively. Thus, endocytosis dysfunction may be an underlying cause of ALS pathology.

---

[1] Department of Molecular and Cellular Biology, University of Arizona, Tucson, AZ 85721, USA. [2] Departments of Neuroscience and Neurology, University of Arizona, Tucson, AZ 85721, USA. Correspondence and requests for materials should be addressed to J.R.B. (email: rbuchan@email.arizona.edu)

ALS is a neurodegenerative disease characterized by motor neuron loss, muscle weakness, paralysis, and eventually respiratory failure. Death typically occurs within 3–5 years post diagnosis[1]. Most ALS cases are sporadic with only 5–10% of cases being inherited. No effective treatment for ALS currently exists.

A hallmark of ALS is the accumulation of cytoplasmic ubiquitinated protein aggregates in affected motor neurons, of which TAR DNA-binding protein 43 (TDP-43) is the primary component[2]. Normally, TDP-43 is a nuclear protein with many roles in messenger RNA (mRNA) metabolism, but in ALS, TDP-43 relocalizes to cytoplasmic aggregates. Several studies suggest these TDP-43 aggregates induce disease partly via a toxic gain of function. However, the source of such toxicity, and possible loss of function (LOF) effects remain highly debated[3]. Regardless,

there is significant interest in understanding how TDP-43 aggregates form, and in developing therapeutic strategies that prevent TDP-43 aggregation or facilitate TDP-43 aggregate clearance.

TDP-43 aggregates may form due to TDP-43 concentration in cytoplasmic stress granules (SGs)[4]. SGs are dynamic, membrane-less assemblies composed of non-translating mRNA–protein (mRNP) complexes[5]. Interestingly, many genes that are mutated in ALS include RNA binding proteins that also localize in SGs (e.g., TDP-43, FUS, hnRNPA1, Ataxin-2[6]). Such mutations usually make these proteins more aggregation prone, which can lead to more rapid or persistent SG assembly[7]. Supporting a role of SGs in TDP-43 toxicity, preventing SG assembly either genetically or chemically has shown therapeutic benefit in models of TDP-43 toxicity[8, 9].

**Fig. 1** TDP-43 toxicity and turnover in yeast are only modestly affected by autophagy. **a** Turnover of TDP-43-YFP protein in WT and null strains in mid-log phase. Time indicates period following *GAL1*-transcriptional shut off. TDP-43-YFP protein level was quantified relative to a Pgk1 loading control. **b** Vacuolar degradation of TDP-43-YFP in WT cells in mid-log phase, +/−4 h nitrogen starvation (N-S), assessed by accumulation of free YFP fragment. Protein levels quantified as above. **c** Turnover of TDP-43-YFP in WT cells in mid-log phase, +/−N-S. Time indicates period following CHX addition (0.2 mg/ml). Protein levels quantified as above. **d** WT strain was transformed with TDP-43-mRuby2 and GFP-Atg8. % value indicates co-localization of TDP-43 foci with Atg8. Scale bar = 2 μm. **e** Indicated strains transformed with TDP-43-YFP or Pab1-GFP were harvested and vacuolar free YFP/GFP was detected. Protein levels quantified as above. **f** Turnover of TDP-43-YFP indicated strains (vacuolar protease mutants) examined as in **c**. **g** Serial dilution growth assay of WT and indicated isogenic null strains expressing vector or TDP-43-YFP

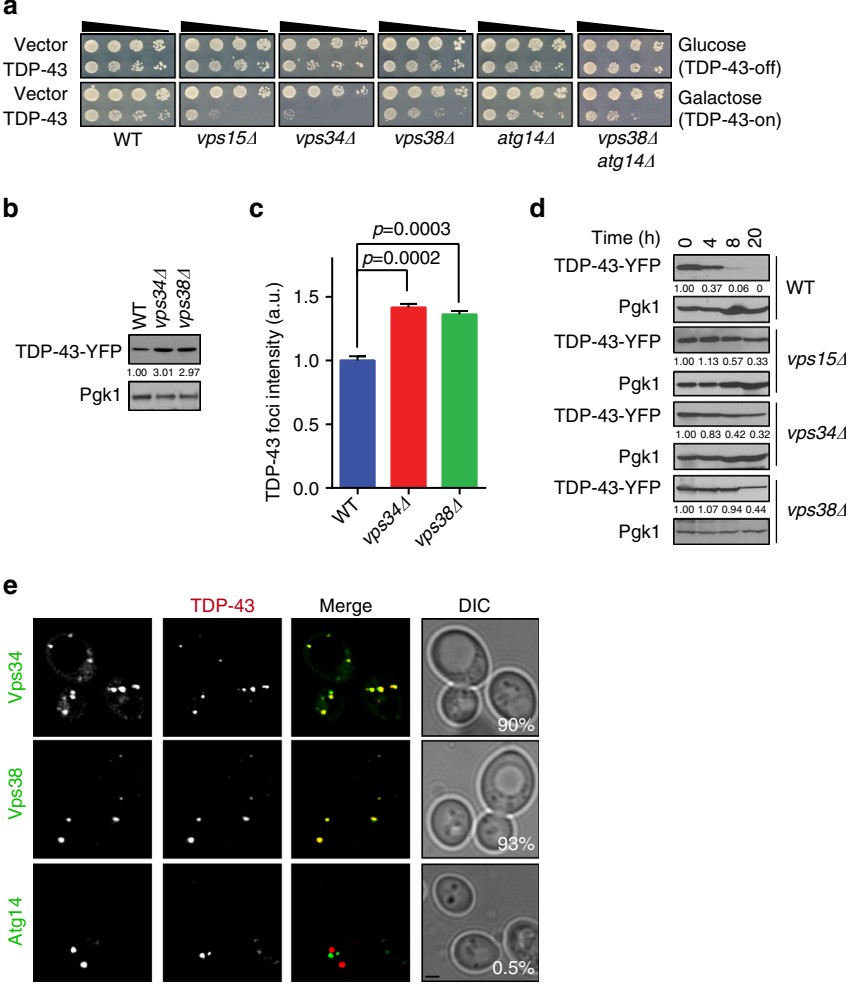

**Fig. 2** Yeast PI3K complex II affects TDP-43 toxicity and turnover. **a** Serial dilution growth assay of indicated isogenic null strains expressing vector and TDP-43. **b**, **c** Indicated strains expressing TDP-43-YFP were examined/quantified for overall TDP-43-YFP levels as in Fig. 1e (**b**) or foci intensity (**c**). Significance was assessed by one-way ANOVA. Repeated data are shown as mean ± s.e.m. **d** TDP-43-YFP turnover in indicated strains, as in Fig. 1a. **e** WT strain co-transformed with TDP-43-mRuby2 and Vps34-GFP, Vps38-GFP, or Atg14-GFP. % value indicates co-localization of TDP-43 foci with Vps34, Vps38, or Atg14. Scale bar = 2 µm

In contrast, other studies suggest that impaired turnover of TDP-43, either via defects in autophagy or proteasomal function, contributes to ALS pathology[10, 11]. Mutations in genes involved in either process such as UBQLN2, OPTN, p62, and VCP occur in some ALS patients[6]. Furthermore, induction of non-selective autophagy rescues TDP-43 toxicity in human iPSC-derived neurons, astrocytes, and a mouse frontotemporal lobar dementia model[12, 13].

Interestingly, SGs are targeted by an autophagic pathway called granulophagy[14], the mechanism of which is not yet fully understood. Evidence that granulophagy is selective includes a dependence on "adaptor" (Atg11, yeast) and "receptor" (p62, human)[15] proteins dispensable for non-selective autophagy. Additionally, granulophagy is active only under specific stress conditions, or when cytoplasmic mRNA decay is blocked in yeast, which causes aberrant SG formation. Since TDP-43 localizes in SGs, we initially hypothesized that inducing granulophagy may help clear TPD-43 aggregates.

For many neurodegenerative diseases featuring protein aggregates, yeast has helped identify relevant modifiers of disease aggregate toxicity[16]. For example, TDP-43 expression in yeast, which recapitulates TDP-43 aggregation, SG localization, and cellular toxicity, enabled the identification of Pbp1 as a modifier of TDP-43 toxicity. This lead to the discovery of Ataxin-2 (Pbp1 ortholog) in humans as an ALS risk factor[17].

Here, we discover that granulophagy and non-selective autophagy weakly affect TDP-43 turnover, and do not affect TDP-43 toxicity in yeast. Instead, TDP-43 turnover, aggregation, and toxicity strongly depend on endocytosis, both in yeast and human cell lines. TDP-43 inhibits endocytosis, and co-localizes with endocytic proteins in yeast, cell lines, and ALS frontal cortex tissue. Additionally, endocytic defects exacerbate motor neuron dysfunction in a TDP-43 ALS fly model. Importantly, enhancing endocytosis increases TDP-43 turnover, and reduces TDP-43 toxicity and motor neuron dysfunction. Our work implicates defective endocytosis function in ALS pathology.

## Results

**Effect of autophagy on TDP-43 turnover and toxicity in yeast.** Given prior studies on TDP-43 turnover and our interest in granulophagy, we examined how non-selective autophagy, granulophagy, and the proteasome affected TDP-43 turnover in yeast. TDP-43 was expressed via a galactose-regulatable promoter. Galactose concentrations were optimized to limit overexpression and help identify mutants that exacerbate TDP-43 toxicity (Supplementary Fig. 1a).

We first examined TDP-43-YFP stability in atg1Δ and atg8Δ strains, which lack genes essential for autophagic pathways

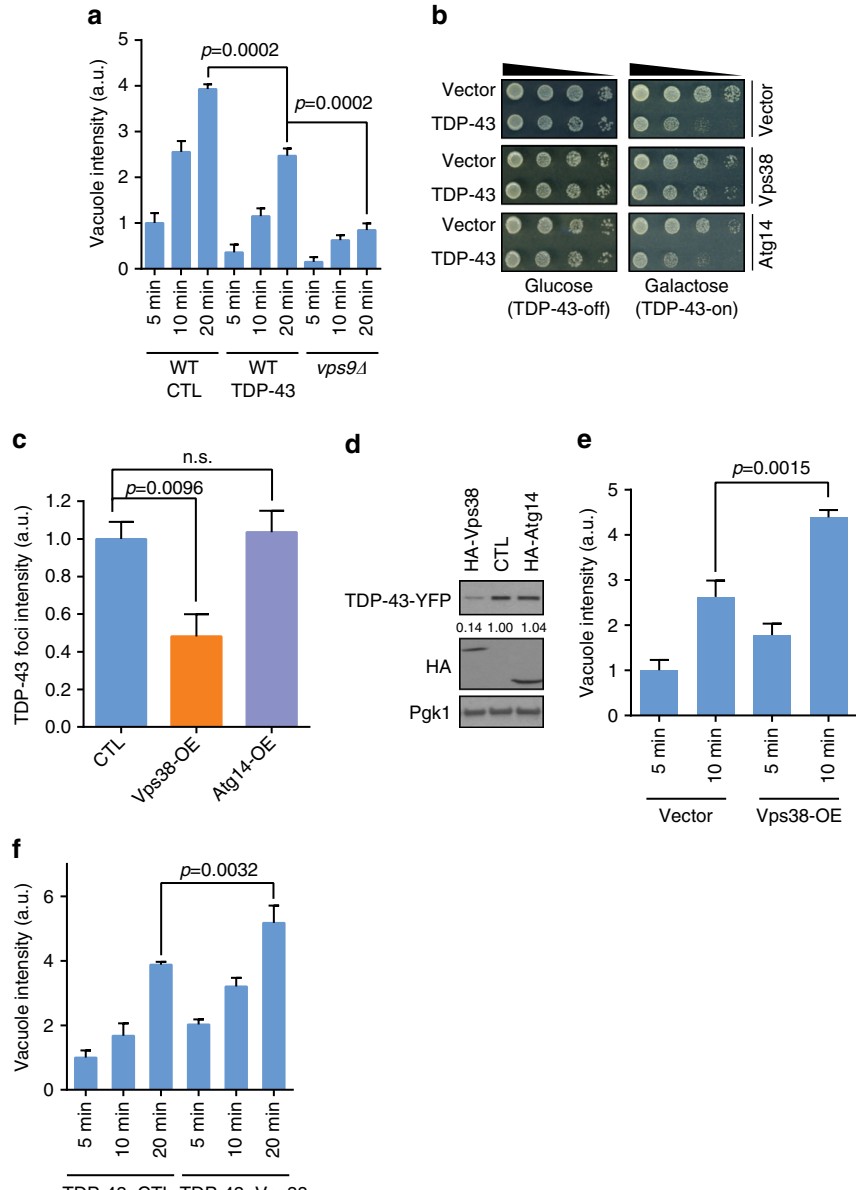

**Fig. 3** TDP-43-induced toxicity and endocytosis defects are rescued by Vps38 overexpression in yeast. **a** WT cells transformed with vector or TDP-43-YFP, and *vps9Δ* cells were stained with FM4-64 dye (8 μM) for indicated time and examined. Vacuolar membrane staining intensity was analyzed. Significance was assessed by one-way ANOVA. **b** WT co-transformed with TDP-43-YFP and either multi-copy empty vector, Vps38, or Atg14-expressing plasmids. **c**, **d** Same strains were quantified for TDP-43-YFP foci intensity (Supplementary Fig. 3) (**c**) and protein levels (**d**), which were assessed relative to Pgk1 loading control. Significance in **c** assessed via one-way ANOVA. n.s. = no significance. Repeated data are shown as mean ± s.e.m. **e**, **f** WT cells transformed with indicated plasmids were examined as in **a**. Significance was assessed by two tailed Student's *t* test. Data are shown as mean ± s.e.m

including granulophagy; note that no difference in TDP-43 stability was observed with or without a YFP tag (Supplementary Fig. 1b). During log phase, TDP-43-YFP stability was barely affected in *atg1Δ* or *atg8Δ* cells (Fig. 1a), consistent with autophagy usually being inactive during log phase. Proteasomal inhibition also had no effect on TDP-43-YFP stability in log phase (Supplementary Fig. 1c). Blocking autophagy and proteasomal function also showed only minor effects on TDP-43 stability (Supplementary Fig. 1d). This suggested the existence of an alternative TDP-43 turnover mechanism under normal growth conditions.

We next examined if inducing non-selective autophagy by nitrogen starvation ("N-S") promoted TDP-43 turnover by use of a steady-state YFP/GFP fragment assay, in which appearance of a YFP or GFP fragment indicates vacuolar turnover of YFP or GFP-fusion proteins[18, 19]. YFP fragment levels indeed increased in TDP-43-YFP expressing cells undergoing N-S stress (Fig. 1b). N-S stress also decreased TDP-43 stability (Fig. 1c) and TDP-43 foci intensity (Supplementary Fig. 1e). These data argue that inducing non-selective autophagy promotes TDP-43 turnover.

To determine if TDP-43 is subject to granulophagy, we first examined if TDP-43-YFP localizes in yeast SGs, identified by Poly (U) binding protein 1 (Pub1) foci. Pub1, and its mammalian ortholog TIA-1, are SG assembly proteins[5]. TDP-43 indeed localized in yeast SGs during stationary phase, when granulophagy is active[14] (Supplementary Fig. 1f). This mirrors previous observations[8], albeit we also observed small SG-independent TDP-43 foci. Notably, cycloheximide (CHX) treatment, which

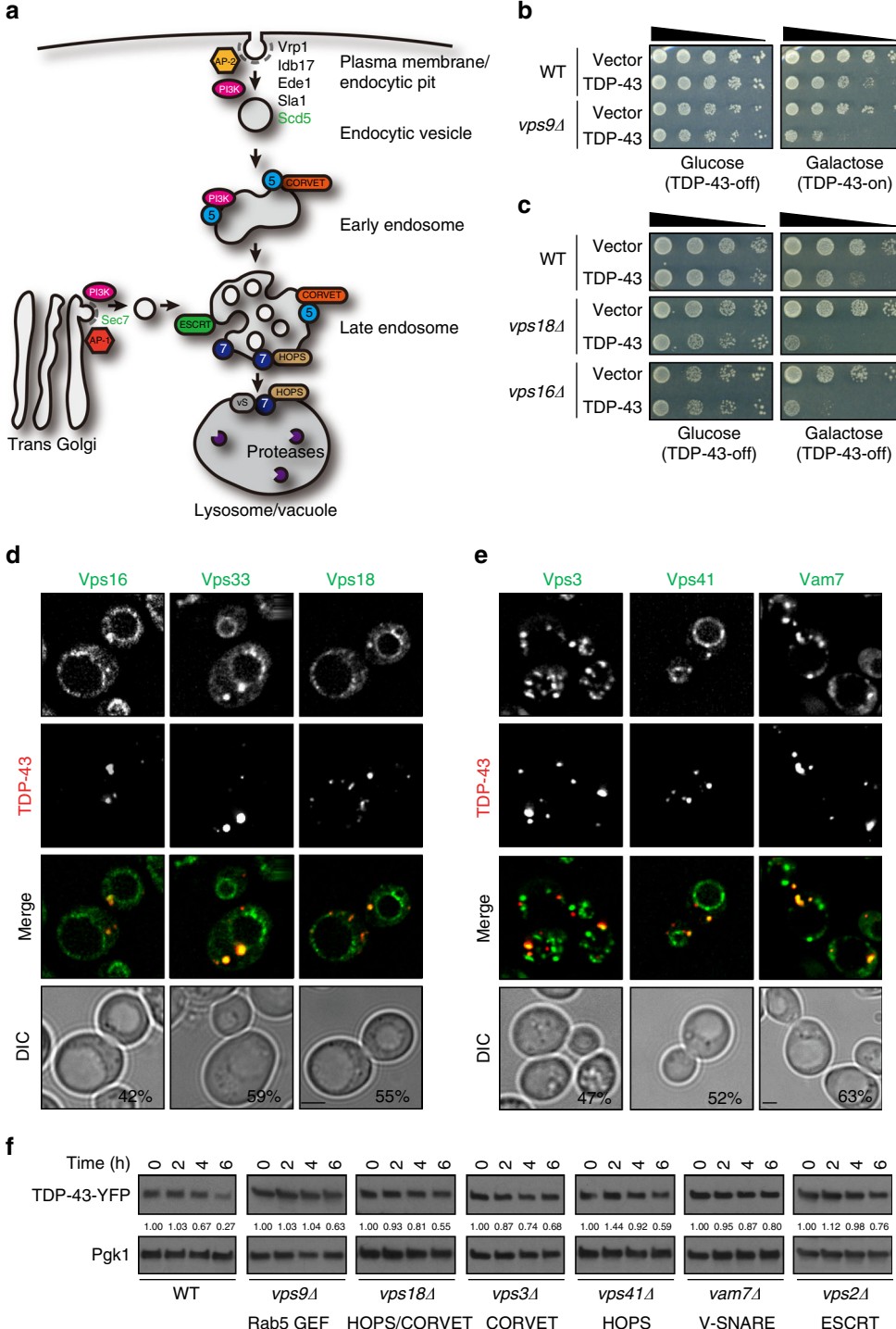

**Fig. 4** Endocytosis affects TDP-43 toxicity and turnover in yeast. **a** Schematic of endocytosis and trans-Golgi vacuolar trafficking, highlighting all genes/complexes tested in yeast; green text (Sec7, Scd5) indicates GFP co-localization experiments only conducted. **b** WT and *vps9Δ* (Rab5 GEF) strains expressing vector or TDP-43-YFP. **c** WT, *vps18Δ*, and *vps16Δ* strains expressing vector or TDP-43-YFP. **d**, **e** Strains expressing indicated GFP-tagged proteins were transformed with TDP-43-mRuby2. % value indicates co-localization of TDP-43-mRuby2 foci with GFP-tagged proteins. Scale bar = 2 μm. **f** Turnover of TDP-43-YFP in WT and indicated null strains; conducted as in Fig. 1c

disassembles SG, caused loss of Pub1 but not TDP-43 foci (Supplementary Fig. 1g, h). This supports a prior suggestion that SGs may be dispensable nucleation sites for TDP-43 aggregation[4].

We next examined if TDP-43-mRuby2 aggregates co-localized with an autophagsomal marker, GFP-Atg8 (Fig. 1d), and other selective autophagy proteins (Atg11-GFP, Atg19-GFP; Supplementary Fig. 1i). TDP-43 foci modestly co-localized with Atg8

and Atg19, but not Atg11, which facilitates granulophagy[14]. Next, we examined if TDP-43 turnover resembled that of poly(A) binding protein 1 (Pab1), which is a core SG protein that binds mRNA poly(A) tails and is a protein target of granulophagy[14]. Interestingly, following Dcp2 inactivation in a temperature sensitive *dcp2-7 atg15Δ* strain, which blocks mRNA decay and potently induces granulophagy[14], Pab1-GFP accumulated in

vacuoles, whereas TDP-43-YFP remained mostly in cytoplasmic foci (Supplementary Fig. 1j). Additionally, during stationary phase, vacuolar turnover of Pab1-GFP required core autophagy genes (Fig. 1e). In contrast, TDP-43-YFP turnover was not fully reliant on core autophagy genes for vacuolar turnover (Fig. 1e). Similar differences in TDP-43 and Pab1 turnover were also observed when non-selective autophagy was induced by N-S stress (Supplementary Fig. 1k). Finally, vacuolar turnover of TDP-43 was confirmed by examining TDP-43 stability in *prc1Δ*, *prb1Δ*, *cps1Δ*, and *pep4Δ* vacuolar protease mutants during log phase. TDP-43 stability increased to varying degrees, likely reflecting vacuolar protease redundancy (Fig. 1f). Collectively, these data indicate that TDP-43 in yeast is not a granulophagy target, but can be partially cleared by non-selective autophagy, particularly under strong inducing conditions. Importantly, TDP-43 was also degraded in vacuoles via a non-autophagic pathway.

Finally, deletion of core and granulophagy-specific autophagy genes did not affect TDP-43 toxicity (Fig. 1g; Supplementary Fig. 1l). As a control, we verified enhanced toxicity of TDP-43 in *tif4631Δ* as previously described[20]. Thus, blocking autophagy does not alter TDP-43 toxicity in yeast.

**PI3K complex mutants alter TDP-43 toxicity and turnover.** Two non-autophagic vacuolar trafficking pathways that could promote TDP-43 turnover include endocytosis and Golgi vacuole trafficking[21]. Both require the conserved class III phosphoinositide 3-kinase (PI3K) Vps34. Vps34 functions in two conserved PI3K complexes, "I" and "II", which include Vps15 (a Vps34 regulatory S/T kinase), Vps30/Beclin 1, and either Atg14 (complex I-specific) or Vps38/UVRAG (complex II-specific). Yeast PI3K complex I promotes autophagy, whereas PI3K complex II promotes endocytosis and Golgi vacuole trafficking[22]. In mammalian cells, conflicting evidence exists as to whether PI3K complex II also aids autophagy besides promoting endocytosis[23, 24]. Regardless, PI3K inhibition induces increased death of iPSC-derived motor neurons that express a TDP-43 ALS disease variant[25]. Examining PI3K mutants was thus of interest.

Importantly, both *vps34Δ* and *vps15Δ* strains showed enhanced sensitivity to TDP-43 expression (Fig. 2a). To determine if PI3K complex I or II differentially affected TDP-43 toxicity, *vps38Δ* and *atg14Δ* mutants were examined. Only the *vps38Δ* strain showed some sensitivity to TDP-43 expression (Fig. 2a), albeit less than in *vps34Δ* and *vps15Δ* strains. This preliminarily suggested that the PI3K complex II was more important in affecting TDP-43 toxicity. We also examined a *vps38Δ atg14Δ* double mutant to theoretically inhibit function of both PI3K complexes. However, TDP-43 toxicity in the double mutant was still less than in the *vps34Δ* and *vps15Δ* strains, suggesting that the Vps34 kinase may retain some functional activity even in the absence of subunits (Vps38, Atg14) thought to be core to PI3K complexes. As expected, re-introduction of Vps15, Vps34 and Vps38 expression plasmids into their respective null strain backgrounds suppressed TDP-43 toxicity (Supplementary Fig. 2a). However, introduction of a Vps15 E200R mutant, which cannot activate Vps34 kinase activity, failed to suppress TDP-43 toxicity in the *vps15Δ* background (Supplementary Fig. 2b). In summary, Vps34 kinase activity, likely within PI3K complex II, helps mitigate TDP-43 toxicity.

We next examined if *vps34Δ* and *vps38Δ* mutants affected TDP-43 foci and protein levels, and indeed both were significantly increased (Fig. 2b, c; Supplementary Fig. 2c). TDP-43 protein stability was also significantly increased in *vps15Δ*, *vps34Δ*, and *vps38Δ* strains (Fig. 2d), more so than effects observed with autophagy and/or proteasomal blocks. Finally, we examined if TDP-43 co-localized with PI3K complex II proteins.

Strikingly, >90% TDP-43 foci co-localized with Vps34 and Vps38, but not with Atg14 (Fig. 2e). In summary, PI3K complex II defects correlate with increased TDP-43 protein stability and foci intensity. TDP-43 foci also strongly co-localized with PI3K complex II, possibly within a PI3K complex II-mediated vacuole trafficking pathway.

**TDP-43 expression inhibits endocytosis in yeast.** Some neuro-degenerative disease aggregates inhibit endocytosis[26], though this observation has not previously been made concerning TDP-43. Given this, and our PI3K data, we were curious if TDP-43 expression may impair endocytosis in yeast.

We measured endocytosis rate using an FM4-64 assay[27], in which extracellular FM4-64 dye is endocytosed and stains vacuolar membranes. As a control for endocytic dysfunction, we included a *vps9Δ* strain, which encodes a guanine nucleotide exchange factor (GEF) for Rab5 proteins (Vps21/Ypt52/Ypt53) in yeast. Lack of Rab5 function causes a strong block in endocytosis between early (EE) and late endosome (LE) formation; see ref. [28] for review of all Rab protein functions.

Importantly, in TDP-43 expressing cells, endocytosis was significantly inhibited relative to WT (Fig. 3a; Supplementary Fig. 3a). We obtained equivalent results with a similar assay, in which the dye lucifer yellow is endocytosed and accumulates within vacuoles[29] (Supplementary Fig. 3b). These data are significant as defective endocytosis could be a cause of TDP-43 toxicity in disease, and explain why TDP-43 foci intensity and protein levels increase in class II PI3K mutants (Fig. 2b–d).

**Enhancing endocytosis reduces TDP-43 levels and toxicity.** To further examine if PI3K complex I and II had differential effects on TDP-43 toxicity and turnover, we examined the effect of overexpressing Atg14 (I subunit) or Vps38 (II subunit) in TDP-43 expressing yeast. Strikingly, TDP-43 toxicity was almost completely rescued by Vps38 overexpression, whereas Atg14 overexpression had no effect (Fig. 3b). In addition, Vps38 overexpression strongly reduced TDP-43-YFP foci intensity and protein levels (Fig. 3c, d; Supplementary Fig. 3c). Intriguingly, Vps38 overexpression also enhanced the rate of endocytosis (Fig. 3e; Supplementary Fig. 3d), and suppressed TDP-43-induced defects in endocytosis rate (Fig. 3f; Supplementary Fig. 3e). To our knowledge, this is the first description of Vps38 overexpression driving endocytosis. Although the mechanism of this is unclear, Vps38 may be rate-limiting for PI3K II complex formation. Regardless, these data suggest a simple model in which enhancing endocytosis by Vps38 overexpression counteracts TDP-43 toxicity by stimulating endocytic TDP-43 turnover.

**Endocytosis affects TDP-43 toxicity and turnover in yeast.** To fully distinguish whether TDP-43 toxicity is reliant on PI3K-mediated endocytosis or Golgi vacuole trafficking pathways (Fig. 4a), we examined TDP-43 toxicity in strains whose deleted genes act uniquely in either process. Importantly, yeast lacking genes that function in early endocytosis steps at the plasma membrane, including *VRP1, IDB17, EDE1, SLA1,* and *APL3* (AP-2 subunit), showed enhanced TDP-43 toxicity (Supplementary Fig. 4a). In contrast, the absence of Aps1 and Apm1 (both AP-1) subunits, which promote clathrin-dependent transport of trans-Golgi network (TGN) vesicles to LEs, exhibited no effects on TDP-43 toxicity (Supplementary Fig. 4a). Clathrin mutants (*chc1Δ*) showed high sensitivity to TDP-43 expression (Supplementary Fig. 4a), albeit clathrin aids both endocytosis and Golgi vacuolar trafficking. TDP-43 also partially localized with endocytosis proteins Apl1, Apl3 (AP-2 subunits), and Scd5 (aids

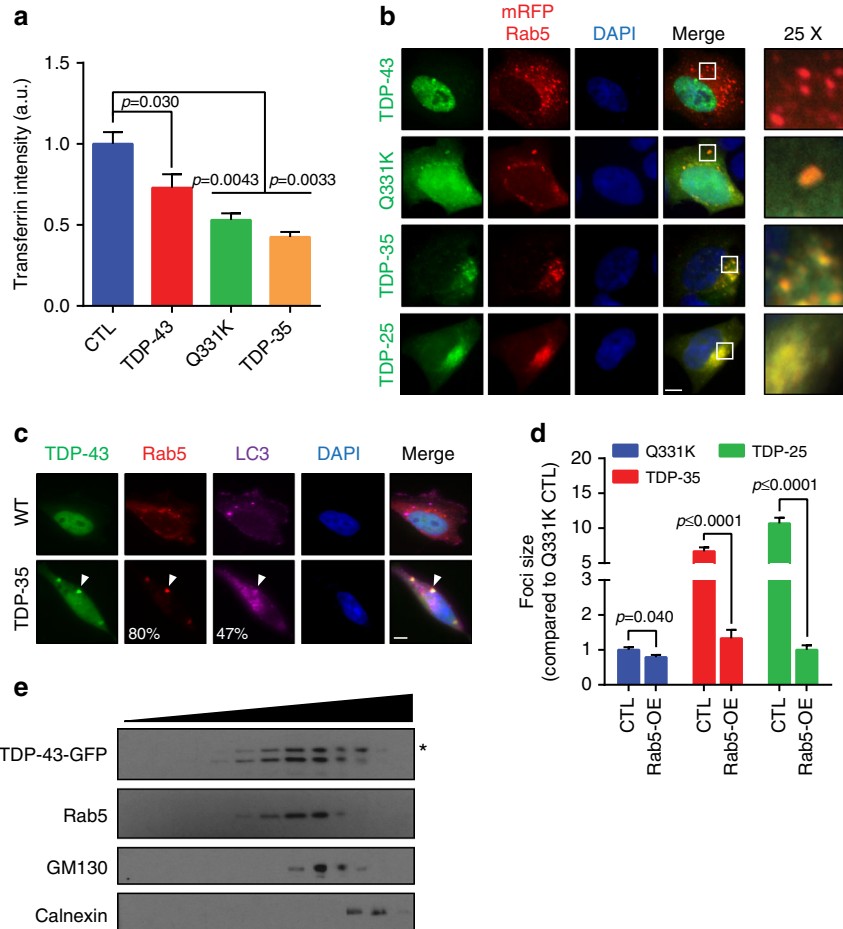

**Fig. 5** TDP-43 foci co-localize with human cell endosomes. **a** HEK293A cells transfected with vector, TDP-43-, TDP-43 Q331K-, and TDP-35-GFP plasmids were stained with transferrin-633 and endocytosis rate was determined by cellular incorporation (Supplementary Fig. 6a). Significance was assessed by one-way ANOVA. Repeated data are shown as mean ± s.e.m. **b** HEK293A cells were transfected with Rab5-mRFP and either TDP-43-, TDP-43 Q331K-, TDP-35-, or TDP-25-GFP plasmids and examined via immunofluorescence. Scale bar = 5 μm. **c** Immunofluorescence was performed in HEK293A cells transfected with TDP-43-GFP and TDP-35-GFP. The percentage of TDP-43 foci that co-localized with endogenous Rab5 or LC3 foci was calculated. Arrowhead indicates TDP-43 co-localization with Rab5 and LC3. Scale bar = 5 μm. **d** Comparison of TDP-43 foci size +/− Rab5 overexpression. Significance was assessed by two tailed Student's *t* test. **e** Organelle fractionation was performed in TDP-43-transfected HEK293A cells. Detection of endogenous Rab5, GM130, and Calnexin indicate EE, Golgi, and ER components. The asterisk indicates full-length TDP-43-GFP

endocytic vesicle formation[30]; Supplementary Fig. 4b). In contrast, Sec7, a GEF that promotes Golgi vacuole trafficking and AP-1 recruitment[31], exhibited no co-localization with TDP-43 (Supplementary Fig. 4b). These data indicate that PI3K complex II effects on TDP-43 reflect its role in endocytic trafficking, and that TDP-43 localizes with proteins acting at early endocytic pathway steps.

We next examined co-localization of TDP-43 with later steps in endocytosis (Fig. 4a). Rab5 proteins (Vps21, Ypt52) are markers of EEs and to some extent LEs. Rab7 (Ypt7) typically localizes to LEs and the vacuole. Partial co-localization of TDP-43 with Rab5/7 proteins was observed (Supplementary Fig. 4c), suggesting that TDP-43 localizes at multiple endocytosis steps.

Given Rab5 redundancy, we examined TDP-43 toxicity in a *vps9Δ* strain (Rab5 GEF). Notably, *vps9Δ* strains showed a strong sensitivity to TDP-43 expression, even accounting for modest sickness of the *vps9Δ* strain alone (Fig. 4b). *Vps9Δ* strains also showed increased TDP-43 protein stability and foci intensity (Fig. 4f; Supplementary Fig. 5a, b).

We next examined the CORVET complex, which facilitates EE fusion, and the HOPS complex, which facilitates fusion events at the vacuole/lysosomal membrane[32]. Both complexes share four

core proteins (Vps11, Vps16, Vps18, and Vps33), two of which (Vps16 and 18) increase TDP-43 toxicity when absent (Fig. 4c), and co-localize with TDP-43 foci, as does Vps33 (Fig. 4d). *Vps16Δ* and *vps18Δ* strains also showed increased TDP-43 foci (Supplementary Fig. 5a, b). Unique CORVET (Vps3, Vps8) and HOPS complex members (Vps39, Vps41) also exhibited increased TDP-43 toxicity when absent (Supplementary Fig. 5c). In addition, Vps3 and Vps41 showed co-localization with TDP-43 (Fig. 4e), with *vps3Δ* yeast also showing increased TDP-43 foci (Supplementary Fig. 5a, b). Finally, TDP-43 protein stability in core (*vps18Δ*), CORVET (*vps3Δ*), and HOPS-specific (*vps41Δ*) mutants was also increased (Fig. 4f).

We also examined ESCRT complex components that facilitate LE biogenesis and membrane invagination[33]. Notably, TDP-43 toxicity increased in *vps27Δ* (ESCRT-0), *vps28Δ* (ESCRTI), *vps36Δ* (ESCRT-II), and *vps2Δ* (ESCRTIII) strains (Supplementary Fig. 5d). Additionally, a *vps2Δ* strain exhibited slower turnover of TDP-43-YFP (Fig. 4f) and increased TDP-43 foci (Supplementary Fig. 5a, b). We also examined a *vps4Δ* strain, as this AAA-ATPase facilitates ESCRTIII complex recycling, LE membrane invagination, and lysosomal sorting[34]. Vps4 is also transcriptionally repressed by TDP-43 in neurons[35]. We observed

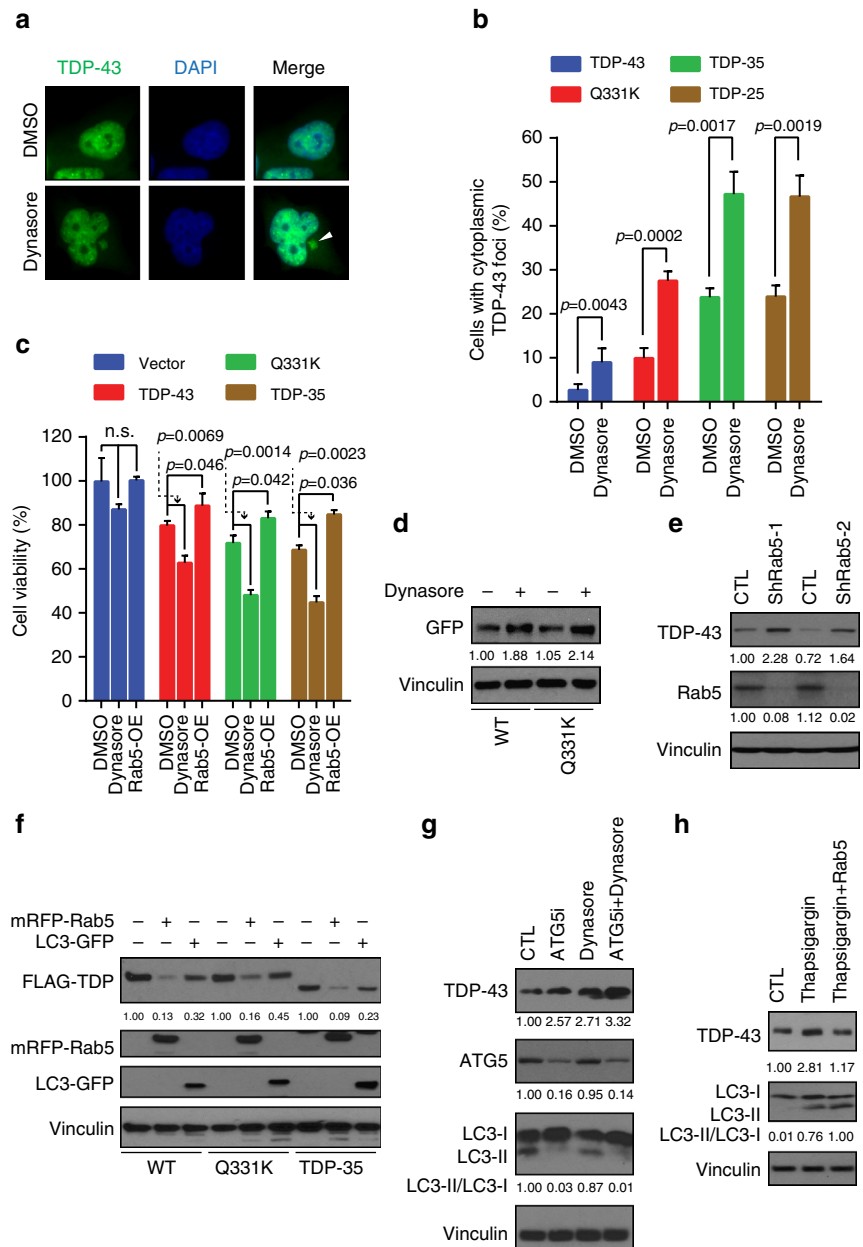

**Fig. 6** Endocytosis and autophagy affect TDP-43 degradation. **a**, **b** HEK293A cells were transfected with TDP-43 WT-, TDP-43 Q331K-, TDP-35-, or TDP-25-GFP and treated with 40 μM dynasore for 24 h. Immunofluorescence was performed; TDP-43-GFP shown in **a**. Arrowhead indicates TDP-43 cytoplasmic foci. Scale bar = 5 μm. Percentage of cells with cytoplasmic foci quantified in **b**. Significance was assessed by two tailed Student's *t* test. Repeated data are shown as mean ± s.e.m. **c** Stable HEK293A cell line including vector, WT, Q331K, and TDP-35 were constructed, treated with or without 40 μM dynasore or Rab5 overexpression and cell viability was compared after 48 h. Significance was assessed by one-way ANOVA; n.s. = no significance. Data are shown as mean ± s.e.m. **d** HEK293A cells were transfected with TDP-43 WT and Q331K-GFP plasmids, treated with or without 40 μM dynasore for 48 h, then harvested and subject to western blotting and quantification of TDP-43 levels relative to a Vinculin loading control. **e** Endogenous TDP-43 expression level was detected and quantified in control (CTL) and Rab5 knock down cells as above. **f** Rab5 or LC3 was co-transfected with TDP-43 WT or Q331K plasmids. Cells were harvested after 48 h culture and subject to western blotting and quantified as above. Note, higher-molecular weight bands in mRFP-Rab5 row, TDP-35 set (1st and 3rd column) are FLAG-TDP-35. **g** HEK293A cells were treated with ATG5 siRNA knockdown (efficiency indicated in second row) and/or 40 μM dynasore for 24 h. Endogenous TDP-43 and LC3 protein levels were examined and quantified as above. **h** HEK293A cells were treated with 0.3 μM thapsigargin for 24 h or combined with Rab5 transfection. Endogenous TDP-43 and LC3 protein levels were examined and quantified as above

Vps4 protein levels decreased 50% in yeast in the presence of TDP-43 (Supplementary Fig. 5e). Additionally, TDP-43 is more stable in *vps4Δ* mutant cells (Supplementary Fig. 5f).

Lastly, we examined vacuolar SNARE proteins, which aid fusion of endocytic vesicles with vacuoles. Notably, *vam7Δ* and *vam3Δ* strains showed increased TDP-43 toxicity (Supplementary Fig. 5c). TDP-43 also co-localized with Vam7 (Fig. 4e) and a *vam7Δ* strain showed increased TDP-43 stability and foci intensity (Fig. 4f; Supplementary Fig. 5a, b).

In summary, multiple endocytosis proteins localize with TDP-43 in yeast and increase TDP-43 toxicity, foci, and protein stability when absent.

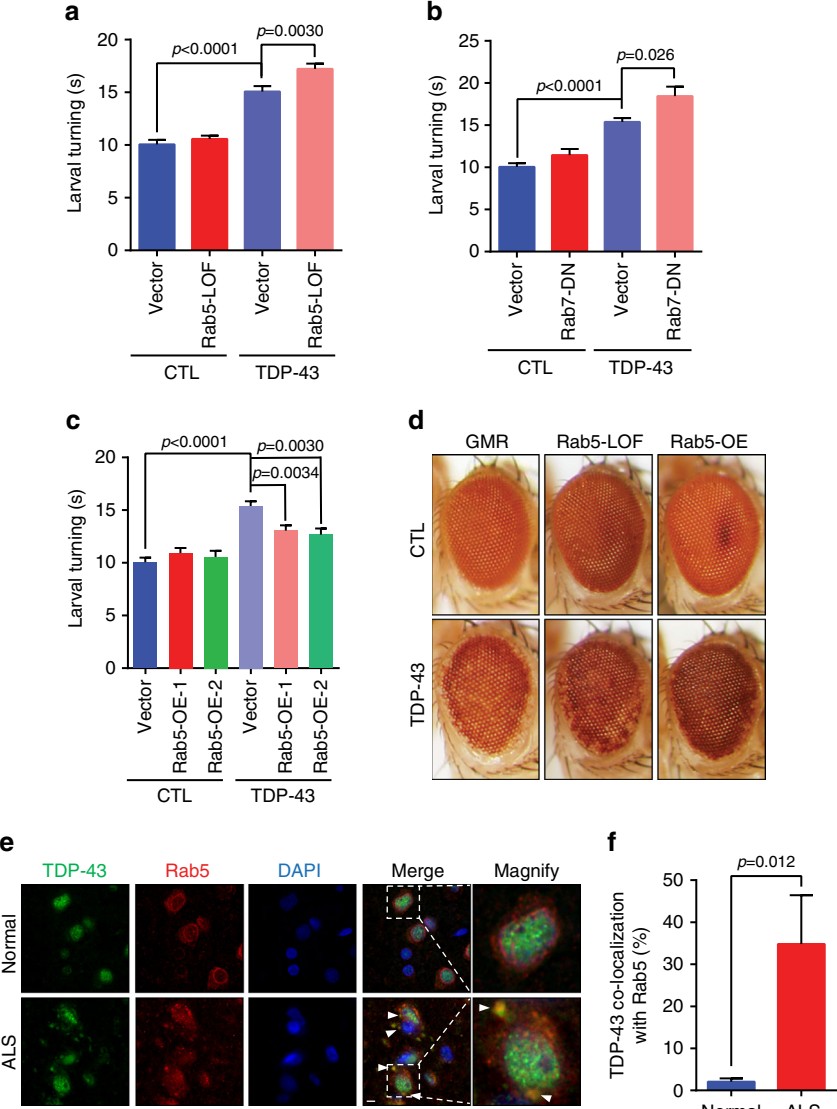

**Fig. 7** TDP-43-induced locomotor dysfunction and neurodegeneration is modulated by key endocytic proteins. **a–c** TDP-43 was specifically expressed in motor neurons +/− distinct Rab mutant backgrounds, and larval turning times (locomotor dysfunction) assessed. **a** Rab5 loss of function (LOF), **b** Rab7 dominant negative (DN), **c** two independent Rab5 overexpression (OE) lines. Significance was assessed by two tailed Student's $t$ test. Data are shown as mean ± s.e.m. **d** Fly retina neurodegeneration assessed by depigmentation; TDP-43 was expressed using GMR GAL4. **e** Fluorescence immunohistochemistry of control and ALS frontal cortex tissue, staining for endogenous TDP-43, Rab5, and DNA (DAPI). Arrowhead indicates TDP-43 co-localization with Rab5 foci. Scale bar = 5 μm. **f** Quantification of TDP-43 cytoplasmic foci exhibiting Rab5 co-localization in **e** five ALS patients and three control patients were examined. Significance was assessed by two tailed Student's $t$ test. Repeated data are shown as mean ± s.e.m

**TDP-43 localizes with endocytic proteins in human cells**. We next examined if TDP-43 was affected by endocytic function in human cells. A complicating issue here is that in human cells, endocytosis and autophagic pathways typically converge prior to lysosome fusion[36]. Additionally, regulators and inhibitors of one pathway can often affect the other (examples discussed below). Nonetheless, we began by transfecting cells with various TDP-43-expressing constructs. These included WT TDP-43, a Q331K disease mutant allele, and aggregation-prone C-terminal fragments of TDP-43 (TDP-25 and 35). These are observed in ALS patient aggregates, and are associated with cellular toxicity[37].

As in yeast, TDP-43 expression, particularly aggregation-prone forms (Q331K, TDP-35), impaired endocytosis as assessed by a transferrin uptake assay (Fig. 5a; Supplementary Fig. 6a; note partial localization of TDP-43 foci with transferrin-labeled endosomes). Additionally, TDP-43 foci strongly co-localized with ectopic and endogenous Rab5, ectopic Vps34, and endogenous UVRAG (Vps38 homolog; Fig. 5b, c, Supplementary Fig. 6b–e). TDP-43 foci also co-localized with endogenous LC3 foci (Atg8 ortholog, autophagosome component; Fig. 5c), supporting prior observations[11]. However, TDP-43 foci co-localized more frequently with endogenous Rab5 foci (Fig. 5c). This suggests a greater association of TDP-43 with endosomal-like compartments over autophagic compartments. Interestingly, the size of Q331K, TDP-35, and TDP-25 foci was significantly reduced in cells overexpressing Rab5 (Fig. 5d; compare Fig. 5b vs. Supplementary Fig. 6b). Multiple studies indicate that Rab5 overexpression enhances endocytosis rates in human cells by stimulating EE fusion and membrane docking events[38, 39], thus this result resembles our yeast Vps38 overexpression data. However, Rab5 overexpression may also enhance autophagy by stimulating autophagosome assembly[40]. Finally, using OptiPrep

gradient fractionation of cellular organelles, TDP-43 and TDP-35 showed preferential association with endosomal fractions over the Golgi and ER (Fig. 5e; Supplementary Fig. 6f). Using sucrose gradient fractionation, clear accumulation of ectopic TDP-43 in both EE and LE fractions, marked by endogenous Rab5 and Rab7 was also detected, with a preference for EE accumulation (Supplementary Fig. 6g). In summary, as in yeast, ectopically expressed TDP-43 inhibits endocytosis and localizes with endocytic (and autophagy) pathway components in human cells.

**Endocytosis affects TDP-43 foci and toxicity in human cells.** We next assessed how inhibiting mammalian endocytosis affected TDP-43 foci levels via use of dynasore, which inhibits dynamin function[41]. Notably, dynasore induced significant increases in TDP-43 foci in cells expressing TDP-43, Q331K, TDP-35, or TDP-25 proteins (Fig. 6a, b).

Next, HEK293A cells stably expressing N-terminal FLAG and C-terminal GFP-tagged versions of TDP-43 (WT, Q331K, and TDP-35) were generated whose expression level approximated endogenous TDP-43 (Supplementary Fig. 7a). The effect on TDP-43 toxicity of inhibiting endocytosis with dynasore or enhancing endocytosis (and perhaps autophagy) by overexpressing Rab5 was then examined in these lines. Although dynasore alone weakly inhibited cell viability, TDP-43-expressing cells showed significant decreases in viability due to dynasore-mediated inhibition of endocytosis (Fig. 6c). In contrast, Rab5 overexpression significantly increased cell viability in all TDP-43-expressing cells. As a control for Rab5 overexpression, we also overexpressed Rab8, which facilitates transport from the TGN and recycling endosomes to the plasma membrane[28]. This did not rescue cell viability in TDP-43 Q331K-expressing cells (Supplementary Fig. 7b), suggesting the Rab5 overexpression rescue effect was specific and likely endocytosis-related.

Thus, as in yeast, defects in human cell endocytosis increases TDP-43 foci formation and toxicity. Conversely, overexpressing Rab5, which stimulates endocytosis (and perhaps autophagy), reduces TDP-43 foci levels and toxicity.

**Endocytosis promotes TDP-43 clearance in human cells.** We next examined if inhibiting endocytosis impaired TDP-43 protein clearance. As in yeast, ectopically expressed TDP-43 protein levels (WT and Q331K) increased when endocytosis was inhibited by dynasore (Fig. 6d). Endogenous TDP-43 levels also increased following stable knockdown of Rab5 or UVRAG (Fig. 6e; Supplementary Fig. 7c).

We next examined how enhancing endocytosis rates by overexpressing Rab5 affected TDP-43 levels, and again utilized Rab8 overexpression as a control. Strikingly, ectopic TDP-43 levels were strongly reduced with Rab5 overexpression (Fig. 6f), whereas Rab8 overexpression had no effect (Supplementary Fig. 7d).

Given prior literature, we also examined the role of autophagy in TDP-43 clearance. As Rab5 can aid autophagosome formation[40], we examined accumulation of lipidated LC3 ("LC3-II"), which is a proxy for autophagosome number and autophagy rate[18]. Relative to rapamycin treatment, which served as a control for autophagy induction, the effect of Rab5 overexpression was subtle (Supplementary Fig. 7e), suggesting that Rab5-mediated effects on TDP-43 clearance may not be driven in large part by autophagic activation. However, we also examined TDP-43 clearance following overexpression of LC3, which drives autophagy[42]. This strongly decreased levels of ectopically expressed TDP-43, albeit not quite as potently as Rab5 overexpression (Fig. 6f). Nonetheless, this data support prior conclusions that autophagy aids TDP-43 clearance in human cells.

To determine if endocytic and autophagic functions can act in parallel in human cells to clear TDP-43 protein, we examined the effect of Atg5 KD (blocks autophagosome formation), dynasore treatment, or a combination of both upon endogenous TDP-43 protein levels. Consistent with prior findings, Atg5 KD indeed caused TDP-43 accumulation, to a similar degree as dynasore treatment (Fig. 6g; lanes 2–3). However, TDP-43 accumulation was greatest with dynasore inhibition in Atg5 KD cells (Fig. 6g; lane 4), indicating that endocytosis and autophagy both contribute to TDP-43 clearance. We also examined the effect of thapsigargin, which blocks autophagy, but not endocytosis, by preventing Rab7 recruitment to autophagosomes[43]. Endogenous TDP-43 levels increased with thapsigargin treatment, consistent with autophagy facilitating TDP-43 turnover (Fig. 6h). Thapsigargin also increased LC3-II levels, consistent with a block of autophagosome fusion with lysosomes (rather than an increase in autophagic rate) as previously observed[43]. Importantly though, Rab5 overexpression, in the presence of thapsigargin, reduced TDP-43 protein levels back to pre-treatment levels. This indicates that a Rab5-enhanced, autophagy-independent mechanism exists that helps clear TDP-43. Collectively, these data indicate there is an autophagy-independent role for endocytosis in TDP-43 clearance in human cells as in yeast.

**Endocytosis affects TDP-43-induced ALS phenotypes in flies.** To test the physiological relevance of our findings, we turned to a TDP-43 *Drosophila* model that recapitulates key ALS features[44]. In this model, WT or disease-associated G298S TDP-43 is specifically expressed in motor neurons, and locomotor dysfunction is assessed by the time a larva takes to re-orientate itself after being turned onto its back. Both WT and G298S TDP-43 expression significantly increases larval turning times relative to controls (Fig. 7a; Supplementary Fig. 8a), indicative of TDP-43-dependent locomotor dysfunction. We next examined how Rab5 LOF, and Rab7 LOF and dominant-negative mutants altered larval turning times with or without TDP-43 or G298S expression. A Rab8 LOF mutant line served as a Rab specificity control. Importantly, larval turning times in flies expressing either TDP-43 allele were significantly increased in all Rab5 and Rab7 mutant contexts, but not in a Rab8 mutant (Fig. 7a, b; Supplementary Fig. 8a–c). Finally, we utilized a qualitative assay of neuronal degeneration in which tissue-specific expression of TDP-43 or G298S in the eye neuroepithelium reveals neurodegeneration as loss of red pigment. This was exacerbated by expression in a Rab5 LOF context (Fig. 7d; Supplementary Fig. 8d). In summary, impairing endocytic protein function increases TDP-43-driven locomotor dysfunction and neurodegeneration in an organismal ALS model.

We next examined if overexpressing endocytic proteins could suppress TDP-43 phenotypes. As in cell lines, Rab5 overexpression increases endocytosis rate in flies[45]. Strikingly, larval turning times were significantly rescued by overexpressing Rab5 (Fig. 7c; Supplementary Fig. 8e) in flies expressing either TDP-43 allele. Similarly, TDP-43-induced eye neurodegeneration phenotypes were suppressed by Rab5 overexpression (Fig. 7d). In contrast, overexpressing Atg8a, which drives autophagic protein clearance in flies[46], had no effect on TDP-43 motor neuron dysfunction (Supplementary Fig. 8f). In summary, increased availability of Rab5 suppresses TDP-43-induced phenotypes in an organismal ALS model.

**TDP-43 co-localizes with Rab5 in ALS patient tissue.** To examine the importance of endocytosis in human ALS, we conducted immunofluorescence staining of TDP-43 and Rab5 in frontal cortex tissue from control or ALS patients. Importantly,

cytoplasmic TDP-43 foci in ALS tissue often co-localized with Rab5 foci, significantly more so than in control tissue, where occasional cytoplasmic TDP-43 foci were observed that rarely co-localized with Rab5 (Fig. 7e, f). Our data therefore suggest that TDP-43 associates with and may be turned over by the endocytosis pathway in ALS patients.

## Discussion

We hypothesize that a significant fraction of TDP-43 is trafficked to vacuoles/lysosomes for turnover via the endocytic pathway (Supplementary Fig. 9). We propose this as in yeast, blocking autophagy and the proteasome had little effect on TDP-43 protein levels or stability under normal conditions (Fig. 1a; Supplementary Fig. 1c, d). However, TDP-43 showed evidence of vacuolar turnover (Fig. 1e), and deletion of vacuolar proteases increased TDP-43 protein levels (Fig. 1f). This indicated that TDP-43 is trafficked to and degraded in vacuoles at least partly via a non-autophagic mechanism. Endocytosis seems to be the mechanism, given stronger co-localization of TDP-43 foci with endocytic proteins over autophagic proteins in yeast and cell lines (Figs. 1d, 2e, 4d, e, 5b, c; Supplementary Figs. 1i, 4b, c, 6b–e), and since TDP-43 co-fractionates with endosomes in human cells (Fig. 5e; Supplementary Fig. 6f, g). Additionally, impairing endocytosis both in yeast (Figs. 2b, d, 4f) and human cell lines (Fig. 6d, e) caused significant accumulation of TDP-43 protein levels, even when autophagy was inhibited (Fig. 6g). Finally, enhancing endocytosis correlated with reduced TDP-43 protein levels in yeast (Fig. 3d, e) and human cells (Fig. 6f), again even when autophagy was inhibited (Fig. 6h).

Other evidence suggests that endocytosis may be perturbed in ALS, and that this may also impact autophagy. First, *FIG4* (mutated in a small percentage of ALS patients[47]) converts PI (3,5)P2 phosphoinositiol to PI(3)P phosphoinositol[48], which aids EE and autophagosome maturation; note PI3K also generates this. Second, *ALS2* is a Rab5 GEF that is mutated in a form of familial ALS[49]. Third, knockdown of ESCRTI (Tsg101) and III (Vps24) proteins results in accumulation of TDP-43 in cytoplasmic foci[50]. Mutations in *CHMP2B*, another ESCRTIII protein, are also associated with rare forms of ALS[51]. Notably, as suggested in studies on ALS2 and the aforementioned ESCRT proteins[50, 52], endocytic dysfunction may also impair autophagy by blocking autophagosome fusion with the endocytic pathway. Fourth, recycling endosome defects that alter signaling at neuromuscular junctions have been observed in a TDP-43 fly model[53]. Finally, *C9ORF72*, the most commonly mutated gene in ALS[6], harbors a DENN domain commonly found in Rab GEF proteins[54], co-localizes with Rab5, and its depletion impairs endocytosis rate in neuroblastoma cells[55]. Conflicting evidence also exists as to whether C9orf72 promotes[56, 57] or inhibits[58] autophagy via interactions with other Rab proteins and/or effects on mTOR signaling and transcription of autophagy/lysosomal genes. Regardless, our data and that of others argues that endocytic dysfunction may be a common mechanism underlying ALS caused by multiple different mutations.

A key question is how TDP-43 traffics with the endocytic pathway, especially as other SG-localizing proteins (e.g., Pab1; Fig. 1e; Supplementary Fig. 1k) depend on autophagy for vacuolar turnover in yeast. What governs differential sorting remains unclear. Regardless, possible mechanisms by which TDP-43 could enter the endocytosis pathway include internalization into LEs via an ESCRT-facilitated process, or entry at the plasma membrane via standard endocytic internalization. Interestingly, TDP-43 can also be secreted[59], which could promote both endocytic entry and intercellular transmission. Alternatively, TDP-43 may disrupt the endocytic pathway such that a hybrid endocytic compartment is formed, which may permit TDP-43 internalization via a novel mechanism. Finally, TDP-43 could traffic on the exterior of endocytic vesicles and be incorporated into vacuoles/lysosomes by an alternative import mechanism.

Another key question is how TDP-43 might inhibit endocytosis. Our FM4-64 data indicate continued presence of FM4-64 at the plasma membrane even at late time points (Supplementary Fig. 3a) in TDP-43 expressing cells. This suggests an early endocytic step may be blocked, though blocks at later endocytosis steps may also occur. As TDP-43 affects multiple steps of gene expression, TDP-43 aggregates could also inhibit endocytosis via a loss of TDP-43 function that alters expression of key endocytic components (e.g., Vps4; Supplementary Fig. 5e, prior studies[35]). Additionally, although knowledge of TDP-43 interaction with specific cellular membranes is lacking, TDP-43 interacts in vitro with membranes via amino acids 311–319, which are critical for its neurotoxicity[60]. Membrane interactions also aid TDP-43 aggregation, particularly mutant forms[60]. This region can also damage membranes[61], which could be related to TDP-43's ability to inhibit endocytosis. Alternatively, TDP-43 aggregates may sequester and inhibit endocytic proteins. One possibility is Rab5, which is strongly concentrated into large TDP-35/25 aggregates when present at endogenous levels (Supplementary Fig. 6b). This sequestration is less evident when Rab5 is overexpressed (Fig. 5b), which correlates with increased TDP-43 clearance (Fig. 6f) and reduced TDP-43 cellular toxicity (Fig. 6c). Interestingly, expression of a polyglutamine huntingtin protein fragment in cell lines also exhibited increased aggregation and toxicity following Rab5 inhibition. Conversely, these phenotypes were reversed by Rab5 overexpression[40]. A second protein of interest is Hsc70, as SOD1 (ALS) and huntingtin protein aggregates can impair endocytosis function partly via Hsc70 chaperone sequestration[26]. TDP-43 aggregates may behave similarly, and may also translationally repress Hsc70 mRNA through a sequestration mechanism, which in turn affects synaptic vesicle cycling[62]. Determining how TDP-43 aggregates inhibit endocytosis, and if this defect is common to other neurodegenerative disease protein aggregates is a key future goal.

Numerous studies have implicated autophagy in the clearance of TDP-43 aggregates[10, 63]. Thus, our finding that endocytosis plays a significant role in determining TDP-43 protein levels, stability, and toxicity may seem surprising. There are many reasons why endocytic contributions to TDP-43 clearance may have been overlooked. First, many studies[10, 11, 63] that inferred autophagic TDP-43 clearance used inhibitors that also inhibit endocytosis (e.g., wortmannin, 3-MA, both inhibit Vps34). In addition, ubiquitination of substrates is usually required for substrate incorporation into LEs via ESCRT protein function[64]. Thus, accumulation of ubiquitinated TDP-43 could indicate endocytic defects, rather than autophagic/proteasomal defects as is commonly assumed. Finally, TDP-43 aggregates do localize with autophagy-associated proteins such as Ubiquilin1, p62, Atg5, and LC3[11, 65] as we too observed (Fig. 5c). However, given that mammalian endocytic and autophagic pathways converge to form amphisomes prior to lysosome formation[36], such co-localization would be expected even for endocytosis-trafficked TDP-43.

However, TDP-43 is cleared by autophagy in yeast when non-selective autophagy is stimulated (Fig. 1b, c; Supplementary Fig. 1e). We also see even clearer evidence for a role for autophagy in TDP-43 clearance in human cells. LC3 foci co-localize with TDP-43 aggregates (Fig. 5c), LC3 overexpression aids clearance of TDP-43 (Fig. 5d), and Atg5 KD and thapsigargin treatment impairs TDP-43 clearance (Fig. 6g, h). Notably though, manipulation of endocytosis has a greater effect than autophagy manipulation on TDP-43 toxicity, turnover, and foci formation in yeast, and on motor dysfunction in flies.

We hypothesize that TDP-43 aggregate clearance may be enacted by both endocytosis and autophagic pathways, or some type of hybrid mechanism given convergence and inter-dependence between the two pathways (Supplementary Fig. 9). The relative contributions of both pathways may vary depending on different cellular environments. However, we suspect that some instances in which autophagy has been implicated as the key effector of TDP-43 aggregate clearance may have ignored a significant role for endocytosis. Further study of endocytosis in the context of ALS promises much in terms of clarifying ALS disease mechanism and identifying novel therapeutic interventions.

## Methods

**Yeast strains and growth conditions**. The genetic background of strains used in this study are listed in Supplementary Data 1. Yeast strains were cultured in YPD medium at 30 °C unless otherwise stated. Strains transformed with plasmids were grown in synthetic defined (SD) medium supplemented with appropriate amino acids. Either 2% glucose or 0.25% galactose plus 1.75% sucrose as a carbon source to inactivate or drive *GAL1*-driven expression of TDP-43 was used unless otherwise stated. Transformation was performed by standard LiAc methods. Strains generated in this study were constructed by homologous recombination methods. Briefly, EUROSCARF antibiotic/biosynthetic cassettes were PCR amplified with 50 bp regions of homology flanking the start and stop codons of the gene to be deleted. These PCR products were transformed into yeast, which were then subject to appropriate selection and PCR conformation to identify yeast in which the gene in question was successfully replaced (Supplementary Data 1).

**Plasmids and reagents**. Details of all yeast and human cell line plasmids are listed in Supplementary Data 1. All plasmids generated in this study were constructed using the Gibson assembly cloning kit (NEB), in accordance with manufacturer guidelines. About 30 bp overlaps were utilized for DNA fragment joining; Sanger sequencing confirmed successful cloning. Chemicals used in this paper: FM4-64 (Thermo Fisher Scientific, T3166), Lucifer yellow (Thermo Fisher Scientific, L453), Thapsigargin (Enzo Life Sciences, BML-PE180-0001), Dynasore (Sigma Aldrich, D7693), MG132 (VWR, 89161-566), CHX (Amresco, 94271-5G), Rapamycin (Fisher Scientific, BP2963-1). ATG5 siRNA was purchased from CST. Short hairpin RNAs were bought from Sigma Aldrich.

**Serial dilution growth assay**. Yeast were cultured to mid-log phase with 0.25% galactose, 1.75% sucrose medium to induce TDP-43 expression. Cells were diluted six-fold serially, spotted onto appropriate selective media plates, and cultured at 30 °C for 2 days before imaging.

**Yeast fluorescence microscopy**. The methods have been described previously[66]. Briefly, stationary phase cells ($OD_{600} > 3.0$) or mid-log cells ($OD_{600}$ 0.3–0.6—all endocytosis rate assay experiments) were examined using a DeltaVision Elite microscope via a ×100 objective. All image analysis was done using Fiji software[67]. TDP-43 foci co-localization with different proteins was scored manually, in a blind manner, in a minimum of 50 cells (in which each cell may contain multiple foci). Cell and foci counts for all microscopy data (including human cell immunostaining) are listed in Supplementary Data 2. Foci intensity was measured using semi-automated thresholding of collapsed Z-stack yeast images to highlight foci; manual inspection ensured no imaging artefacts (e.g., autofluorescent cellular debris) was accidentally scored. All shown images are representative of >3 biological replicates, in which each biological replicate involves analysis of a minimum of 50 cells.

**TDP-43-YFP protein stability assay**. TDP-43 expression in mid-log cells was halted by transcriptional shut off (switching cells to a 2% glucose medium, which halts GAL1 promoter expression). Alternatively, general protein synthesis was halted by adding 0.2 mg/ml CHX to the yeast culture. In both cases, protein lysates were obtained via a standard NaOH method and examined via Western blot.

**Western blotting**. Western blotting was performed via a standard protocol. All images shown are representative of >3 biological replicates. Primary antibodies used are listed in Supplementary Table 1. Full size uncropped blots are shown in Supplementary Fig. 10.

**FM4-64 endocytosis rate assay**. Yeast endocytosis rate assay was conducted as described previously[68]. Briefly, cells were harvested at mid-log phase and kept on ice for 10 min. Then the cells were suspended in YPD medium containing 8 μM FM4-64 at 30 °C for indicated times. Vacuolar membrane intensity was quantified using Fiji.

**Lucifer yellow endocytosis assay**. Lucifer yellow assay was performed as previous reported[29]. Briefly, yeast cells were cultured to the mid-log phase. Cells was concentrated to 5–8 folds in a 1.5 ml tube. Suspend cells in YPD medium with 1 mg/ml Lucifer yellow and incubate for 30 min. Cells were then concentrated and washed three times in PBS with 10 mM sodium azide and 50 mM sodium fluoride. Images were then taken under the microscope.

**Cell culture**. HEK293A cells were cultured in DMEM medium (HyClone, SH30022.01) with 10% FBS (Thermo Fisher Scientific, 26140079) 50 U/ml penicillin and 50 μg/ml streptomycin (Life Technologies, 15070-063). HEK293A cells stably expressing TDP-43, TDP-43-Q331K, TDP-35, and TDP-25 with FLAG at the N-terminus and GFP at the C-terminus were constructed and selected with 1 μg/ml puromycin (Invivogen, ant-pr-1). SH-SY-5Y cells were cultured with 1:1 mixture of Eagle's minimum essential medium (Bio5 Media Service, University of Arizona) and F12 medium (Thermo Fisher Scientific 11330032), 10% FBS.

**Cell line immunofluorescence**. Cells were transfected with plasmids, cultured for 48 h in 8-well chamber slides (Ibidi, 80821), fixed with 4% paraformaldehyde for 15 min and permeabilized with 0.1% Trion X-100 for 5 min. Cells were incubated with indicated primary antibodies overnight at 4 °C after blocking with 5% goat serum. Cells were then incubated with Alexa fluor conjugated secondary antibody (Thermo Fisher Scientific, 1:1000 dilution) for 1 h at room temperature. After three washes with TBS-T, cells were stained with DAPI (Thermo Fisher Scientific, P36931) and imaged using a Deltavision Elite microscope with a ×40 objective. Antibodies used for the immunofluorescence were: HA (BioLegend, 901501, dilution: 1:200), UVRAG (Sigma Aldrich, U7508 dilution: 1:150), Rab5 (CST, 3547, dilution: 1:200), LC3 (CST, 3868, dilution: 1:200). All shown images are representative of >3 biological replicates.

**Sucrose gradient endosome fractionation assay**. Endosomes were fractionated as previously described[68]. Briefly, cells were harvested in homogenization buffer (HB) (250 mM sucrose, 3 mM imidazole, pH 7.4, 1 mM EDTA, 0.03 mM CHX, 1 mM $Na_3VO_4$, 5 mM $Na_4P_2O_7$, 50 mM NaF, 10 mM β-glycerophosphate, protease inhibitor cocktail) and lysed using a dounce glass-glass homogenizer (B-pestle). Gradient columns (Beckman Coulter tubes, 355530) consisted of HB, 25, 35, and 42% sucrose from top to bottom. Gradients were ultracentrifuged at 21,000×$g$ for 3 h at 4 °C. The interfaces between HB and 25% layers, and between the 25 and 35% layers were harvested, which correspond to the late and early endosome-containing fractions, respectively. All samples were examined by western blotting.

**Optiprep gradient organelle fractionation**. The fractionation was performed as previously[69]. Briefly, homogenization of cells was as described for the endosome fractionation assay described above. About 2 ml gradient columns (Beckman Coulter tubes, 355530) were generated with HB buffer and 5–30% OptiPrep with 2.5% increments. Gradients were ultracentrifuged at 100,000×$g$ for 16 h. Each fraction (166 μl; 12 total) was harvested and examined by western blot using antibodies against appropriate organelle markers (Supplementary Table 1 and main text).

**Cell line viability assay**. HEK293A cells expressing empty vector or different stably integrated versions of TDP-43 were cultured in puromycin containing medium for selection purposes. About $1 \times 10^5$ cells were seeded in one well of a six-well plate and treated with or without dynasore (40 μM, Sigma Aldrich, D7693). Cell number was counted after 48 h growth.

**Transferrin-633 endocytosis rate assay**. The transferrin assay was performed in accordance with Thermo Fisher product guidelines. Briefly, cells were kept on ice for 10 min. Then cells were washed with live cell imaging solution (Thermo Fisher Scientific) and then incubated with 25 μg/ml fluorescent transferrin (T23362, Thermo Fisher Scientific) at 37 °C for 15 min. Then cells were washed again with live cell imaging solution, imaged and quantified for intracellular fluorescence using Fiji.

**Drosophila genetics**. All *Drosophila* stocks and crosses were kept on standard yeast/cornmeal/molasses food at 22 °C. Human TDP-43 variants with YFP C-terminal tags were generated as previously described[50]. Untagged TDP-43$^{WT}$ was obtained from J. Paul Taylor and untagged TDP-43$^{G298S}$ was obtained from Takeshi Iwatsubo. Gal4 drivers included the eye-specific GMR Gal4 and the motor neuron driver D42 Gal4. For controls, $w^{1118}$ was crossed with the appropriate Gal4 driver. To manipulate Rab5 $y^1w^*$;P[UASp YFP Rab5]O2, $y^1w^*$;P[UASp YFP Rab5]Pde8$^0$^8b, and $y^{d2}w^{1118}$;P{EPgy2}Rab5EY10619/CyO were used (Bloomington *Drosophila* Stock Center). To manipulate Rab7 $w^{1118}$-;P[UAS Venus Rab7 WT]/CyO, $y^1w^*$-;P[UAS Venus Rab7 CA1]/CyO, $y^1w^*$-;P[UAS Venus Rab7 DN1]/CyO, and $w^{1118}$ Sp/CyO; FRT82B Rab7KO/TM3 were used[70]. To manipulate Atg8a, $y[1$ $w^{[1118]}]$; P{w[ +mC] = UASp-GFP-mCherry-Atg8a}2 was used (Bloomington *Drosophila* Stock Center). To manipulate Rab8, Rab8$^{[1}$ red$^{[1}$ e$^{[4]}$/TM6B, Sb Tb was used (Bloomington *Drosophila* Stock Center).

**Drosophila eye imaging.** Assays were conducted at 25 °C, and performed as described previously[50], using a Leica MZ6 microscope equipped with Olympus DP73 camera.

**Drosophila larval turning assay.** Assays were performed as previously described[50]. Briefly, third instar wandering larvae were placed and acclimated on a grape juice plate at room temperature. The crawling larvae were turned onto their backs (ventral surface facing up), and the time that each larva took to re-orientate (dorsal side facing up), and make a forward movement was recorded. Thirty larvae were used per genotype. Statistical significance was analyzed by Student's *t* test.

**Patient tissue immunofluorescence.** Paraffin-embedded tissue microarray slides (2–5 mm diameter specimens; posterior and superior frontal cortex tissue) were provided by the Southern Arizona VA Healthcare Agency and immunostained using a standard protocol from Abcam (Cambridge, United Kingdom). Briefly, glass slide-adhered samples were dewaxed with xylene and fixed with successive 100, 100, 90, 80, 70% ethanol and distilled water for 10 min each. Antigen retrieval was accomplished by incubating samples in boiled sodium citrate buffer for 20 min. Endogenous peroxidases were quenched with 3% $H_2O_2$ for 30 min. Tissue samples were then blocked with 5% goat serum for 1 h before incubation with primary antibody at 4 °C overnight. Antibody staining of TDP-43 (ProteinTech) and Rab5 (Abcam) was diluted to 1:200 and 1:100, respectively. After four TBS-T washes, samples were incubated with Alexa fluor conjugated secondary antibody at a 1:1000 dilution for 1 h at room temperature. After washing, immunostained tissue samples were mounted with ProLong Gold Antifade DAPI and imaged.

**Data availability.** The data sets generated during and/or analyzed during the current study are available from the corresponding author on reasonable request.

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

## Acknowledgements

We thank Aaron Gitler, Tricia Serio, and Daniel Klionsky for providing various yeast plasmids, and Kun-Liang Guan and Chengyu Liang for providing various mammalian cell plasmids. We thank Torsten Falk for providing SH-SY5Y cells, and the Southern Arizona VA Healthcare Agency for providing us with ALS and control patient tissue. We thank J. Paul Taylor and Takeshi Iwatsubo for provision of untagged WT and G298S TDP-43, respectively. We thank Roy Parker, Hanna Fares, Anita Koshy, May Khanna, Felicia Goodrum, and Tricia Serio for helpful manuscript feedback. This work was supported by funds to J.R.B. from NIH RO1-GM1145664 and the ALS Association, and to D.C.Z. from NIH RO1 NS091299. S.V. and M.C. were supported in part by internal funds from the Provost through the University of Arizona Undergraduate Biology Research Program.

## Author contributions

G.L. performed all yeast and human cell line/tissue experiments, with assistance from F. P. for yeast growth assays; J.R.B. supervised. A.N.C., S.V., and M.C. designed and performed fruit fly experiments; D.C.Z. supervised. G.L. and J.R.B. wrote the paper and analyzed the data, with assistance from A.N.C. and D.C.Z. for fruit fly data interpretation.

## Additional information

**Competing interests:** The authors declare no competing financial interests.

