## [Peer Review File · Nature Communications]

Reviewers' comments:

Reviewer #1 (Remarks to the Author):

Liu and colleagues present data to suggest that endocytosis plays a more important role in regulating TDP-43 levels than autophagy or proteasome function. Their data suggest that modulating endocytosis may be an important therapeutic target for TDP-43 proteinopathies such as ALS and FTD. If true, this would indeed be a novel and significant finding that would change the current model for how TDP-43 levels are regulated. However, their conclusions are based on an artificial overexpression system in yeast, with very limited experiments performed in a *Drosophila* model and in human cells. Thus, I am far from convinced in their main conclusion that "Endocytosis is a key regulator of TDP-43 toxicity and turnover", especially since there is significant overlap between the endocytosis and autophagy machinery—indeed Rab5 and UVRAG, the most significant markers to colocalize with and regulate TDP-43 in human cells in their studies, have also been shown to regulate autophagy. Thus, the authors need to show that their means of "enhancing endocytosis" (e.g. overexpressing Rab5) don't also enhance autophagy (which others have observed). Given how unexpected their findings are, the authors also need to do more work in Neurons and in cells expressing endogenous levels of TDP-43 to confirm their findings under more physiologic conditions.

In Figure 5j, does Rab5-OE rescue reduced viability induced by TDP-43 expression? This appears to be a critical experiment since Rab5-OE dramatically reduces TDP-43 levels in Figure 5h-i. Also, the authors need to show that Rab5 overexpression doesn't upregulate autophagy given that Ravikumar and colleagues have shown that Rab5 modulates mutant huntingtin toxicity through regulation of autophagy (Ravikumar et al J Cell Sci 2008).

The effects of Rab5 and Rab7 inhibition or overexpression on TDP-43 phenotypes in *Drosophila* are quite modest. Others have shown that genes that regulate autophagy are also modifiers of TDP-43 in the fly; since the authors claim that autophagy doesn't play an important role in TDP-43, the authors should compare effects of genetic modification of autophagy. Again, since Rab5 overexpression may also enhance autophagy, I am not convinced that their rescue experiments support their conclusion.

The limited human ALS data are not convincing, particularly given prior reports that TDP-43 does not localize to endosomes (see Matej et al, *Exp Neurol.* 2010;225:133-9. "Increased neuronal Rab5 immunoreactive endosomes do not colocalize with TDP-43 in motor neuron disease.")

Minor points:

The yeast experiments appear well performed and are convincing. However, more details need to be included on the TDP-43-GFP overexpression model. The authors need to show that the GFP tag doesn't affect the turnover of TDP-43. The authors should provide more details on the construct in methods—importantly, is the human TDP-43 3'UTR present? If so, it may be able to autoregulate its mRNA levels which could confound interpretation of protein stability.

Sup fig6 – (a) why is there no colocalization of mutant TDP-43 with transferrin? Colocalization of endogenous Rab5 with transfected TDP-35 and 25 punctae are quite convincing and striking – however, colocalization is much less convincing with overexpression of ALS/FTD-causing TDP-43. At best, they can say that there is "partial" colocalization. In contrast, the colocalization of mutant TDP-43 with endogenous UVRAG is striking. Since UVRAG is implicated in autophagy, might this suggest that these TDP-43 punctae are autophagosomes as has been described in multiple papers? The authors should show autophagosome markers in the same panels—it may be that Rab5 is also mislocalized on autophagosomes in these cells.

"In contrast, TDP-43-GFP only exhibited a modest decrease in

127 vacuolar turnover and no full-length protein accumulation (Fig. 1e)."

It appears to me in Fig 1e that there is a marked decrease in free GFP, suggesting that there is a reduction in trafficking to the lysosome (or decreased lysosome function).

Vps15 E200R effect would be more convincing if they showed Vps15 WT in same panel from same expt.

How does vps38 overexpression "enhance endocytosis"? Has this increase in endocytosis been shown previously?

In discussion, authors argue that there is "emerging evidence suggesting endocytosis is perturbed in als". In the genes and the papers that the authors describe, there is significant overlap between the endocytic pathway and autophagy. For example, emerging evidence suggests c9orf72 plays a role in autophagy rather than "endocytosis".

Reviewer #2 (Remarks to the Author):

In the manuscript entitled "Endocytosis is a key regulator of TDP-43 toxicity and turnover" Liu and colleagues show endocytosis as a new clearance pathway for TDP43. Although there is an effort in this study to include several disease models to support the unique role of the endocytic pathway in rescuing TDP43 proteinopathy there are concerns that need to be addressed prior to publication:

1. The authors claim that inhibition of the proteasome alone or together with autophagy has no effect on TDP43-GFP stability. However, there is an extensive literature that demonstrates how TDP43 is cleared through both the proteasome system and autophagy, including chaperone-mediated autophagy (Wang 2010; Huang 2014, Scotter 2014, Barmada 2014). The authors should resolve this discrepancy by using multiple null strains of the autophagy pathway. Additionally, the authors should inhibit specifically the autophagosome-lysosome fusion with drugs (Eg: Bafilomycin A, ammonium chloride and/or other inhibitors) to further support that autophagy has no effect in clearing TDP43.
2. The total expression levels of TDP43 in the control and TDP43 over-expressing line should be measured and shown.
3. The authors demonstrate that starvation induced-autophagy promotes TDP43 turnover, although the usage of Atg1 and Atg8 mutants do not affect turnover. To the knowledge of this reviewer, non-selective autophagy does involve these Atg proteins. The authors should provide literature to support this data or test multiple null strains and dissect the two autophagy pathways.
4. Fig 1f: the increase of TDP43 in the vacuolar mutants is not convincing. The authors should present quantification of the intensity values
5. Sup Fig 1f: Authors demonstrate that CHX treatment disassembles SG but not TDP43 foci, but the quality of the figure makes the data unconvincing. Images of fluorescence from the single channels and quantification of the intensity values of Pub1 and TDP43 foci might help. Moreover, the intensity of TDP43 foci looks to be reduced with CHX, which does not support the authors statement that TDP43 foci are resistant, so revisions are necessary.
6. Sup Fig 1h. Authors should show Pab1 induction, so data before and after Dcp2 inactivation need to be included.
7. Sup Fig 1i. Intensity values of the full-length GFP Pab1 accumulated in Atg1 and Atg8 mutants needs to be shown.
8. Fig 4f. Intensity values should be included in bar graphs.
9. The authors examine the ESCRT complex components and demonstrate that TDP43 toxicity increases in several strains of mutants for different ESCRT complexes. Interestingly the TDP43 has been recently demonstrated to repress the Vps4b, a known ESCRT disassembly factor (Schwenk 2016). The authors should include this protein in their analysis, provide some insight on how

TDP43 regulates ESCRT, and include this reference in the discussion.

10. The authors show the distribution of TDP43 fragments in Fig 5b. The Rab5 staining in TDP-25 and TDP-35 is not punctate as in the other groups. This looks to be very relevant because of the strong foci reduction observed in figure 5i. Given the importance of TDP43 in recycling endosomes (Schwenk 2016) it is conceivable that these fragments are somehow interfering with the endocytic system. The authors should comment on this observation.

Minor issues

- Improve the image quality of all the figures
- Fig 1d. Add scale bar information
- Sup Fig 1. Add scale bar information
- Sup Fig 3. Include merged figures in every panel
- Fig 5 b. The Rab5-mRFP and DAPI figures are inverted in the TDP-25 fragment
- Line 299. Use the appropriate language (not man but human)
- Sup Fig 6b. Rab5 should not have a nuclear staining. The ICC data need to be improved.
- Sup Fig 6b. Include WT TDP43

Reviewer #3 (Remarks to the Author):

This interesting manuscript examines the contribution of cellular trafficking pathways to the cytoplasmic aggregation of TDP-43, a nuclear RNA binding proteins associated with ALS pathology. Previous findings indicated a link between autophagy and the clearance of TDP-43 aggregates for detoxification. In this manuscript, a yeast system is used that expresses high levels of TDP-43 to generate toxic aggregates. In their survey of yeast mutants, the authors found that endocytosis mutants had more effect on TDP-43 toxicity than mutants disrupting core autophagy-associated pathways. In particular, associations between TDP-43 and homologs of the endosomal Rab5 GTPase were a focus of this paper, though other endosomal components also seemed to affect TDP-43 toxicity in yeast. Validation experiments were performed in flies and human cells to test whether some of the findings in yeast were generally applicable in metazoans.

These are potentially exciting results here that ascribe a possible mechanism to TDP-43 toxicity and by extension, to ALS pathology. An additional plus to this work was the use of three different cellular/organismal models to validate the functional conservation of the findings. Unfortunately, there were several negatives that outweighed these positives. Weak grammar and lack of thorough explanations led to confusion, especially in the description of this work in the context of previous publications. In terms of experiments, some of the results were overstated, and others experiments required comparative controls and clearer reporting of quantification. Specific comments are provided below.

1. The brief introduction in this manuscript provides little explanation of matters important later in the manuscript. To improve readability, the Introduction could meld disjointed and open-ended sentences into single paragraphs with added description. Throughout the text, details are needed to define central terms and concepts. Readers should be told specifically what Pab1 and Pub1 are, as well as any other key protein that is used as a marker or the relevance of any genes that are being tested. Why is Pab1 a "granulophagy target protein?" More to the point, what generally defines "granulophagy" and what makes it a selective autophagy pathway? How is its mechanism different than other autophagy pathways? To inform general readers, many terms in the Introduction and in later sections need better description.

2. At first glance, this paper seems to contradict earlier findings the corresponding author published in a previous "Cell" article (Buchan et al. 2013 153, 1461). In that paper "granulophagy" was shown to "clear" stress granules and P bodies, but in this manuscript the analogous TDP-43 aggregates are relatively unaffected by granulophagy/autophagy. The previous paper used yeast

functional genomics to identify non-essential genes that clear stress granules, but genes involved in endocytosis genes were not a major group identified. This seems at odds with the results reported here. There would be less confusion if the Introduction section thoroughly explained the previous paper and its findings, and related those results to the issue of TDP-43 aggregation in yeast.

3. In the methods the statement "All images shown are representative of >3 biological replicates" was repeated several times. For the protein blots it seems reasonable for the experiments to have been done in triplicate. However, for the microscopy it is unclear what this statement actually means. Presumably more than 3 cells were examined for determining the percentage of co-localization, though there is no mention of exactly how many cells were counted (n = ?). For quantifying fluorescence intensity ("foci intensity") there was no description in the Methods section for how that was done. Moreover, how many foci were counted from how many cells?

4. Fig 1f and Lines 129-131: "...TDP-43 stability in *prc1Δ*, *prb1Δ*, *cps1Δ* and *pep4Δ* vacuolar protease mutants during log phase, wherein TDP-43 stability clearly increased..." It is an overstatement to say that stability "clearly" increased for all the mutants. The effect was minor for *cps1Δ* and *prb1Δ* (*prb1Δ* actually seems to enhance TDP-43-GFP degradation at 3 hours), and subtle differences are not definitive without providing quantification of band intensities for each lane relative to the internal loading control.

5. Line 154-156. "Only the *vps38Δ* strain showed enhanced sensitivity to TDP-43 expression (Fig. 2a), suggesting that PI3K complex II was more important in affecting TDP-43 toxicity." Perhaps as shown in Fig 2a it is not a representative result but there is no major difference in growth comparing the *vps38Δ* mutant with the wild-type control. The sensitivities observed with *vps15Δ* and *vps34Δ* are much more robust. Perhaps the combination of *atg14Δ vps38Δ* would provide a greater growth defect, especially given the pathway redundancy proposed in Fig 7.

6. Fig 3a, Line 183-184. "...in TDP-43 expressing cells, endocytosis was clearly inhibited..." Because FM4-64 internalization assays can be variable, this experiment should include parallel endocytosis mutant controls. The controls would also provide a comparison to show the degree of inhibition. For these assays, why are stationary-phased cells (i.e. not growing) being used? An independent mean(s) of assaying endocytosis would also be appropriate to verify these results (i.e. lucifer yellow uptake, Ste6 uptake and degradation, etc.), in log-phased cells.

7. Fig 3e, Line 195. "...Vps38 overexpression enhanced the rate of endocytosis in TDP-43 expressing cells..." This is a non-result. There is no real difference, and certainly in stationary phased cells such minor differences would be predicted.

8. Fig 4c, Line 233-234. "...three of which (Vps16, 18 and 33) increase TDP-43 toxicity when absent (Fig. 4c)..." Contrary to this statement, *vps33Δ* had no effect on TDP-43 toxicity when compared to wild type.

8. Fig. 5c.... Bands are difficult to see and no specific markers for other compartments are shown (Golgi, microsomes, etc.) to prove that TDP-43 doesn't co-fractionate with everything. Fig 5d.... What is the arrowhead pointing at? Fig 5i.... Why aren't TDP-43 foci quantified? No specificity controls for Rab5 overexpression are shown (Rab6 and/or 8?).

9. Fig 6. The Rab5 and 7 mutants examined did appear to affect TDP-43-induced defects in flies. However, without an examination of other endocytosis mutants, other Rab GTPases as specificity controls, and direct measurements of endosomal transport, it is an overstatement to state: "In summary, increased availability of Rab5 and thus presumably endocytic rate can suppress TDP-43 induced phenotypes in an organismal ALS model."

Minor Comments:

1. Clarity could be enhanced with better grammar; sentence and paragraph structure could be greatly improved. Long sentences with 4 or more clauses should be separated into several clearer sentences. Paragraphs with 1 or 2 hanging sentences could be amalgamated into proper paragraphs with clear topic and concluding sentences.
2. There are minor formatting issues with the References list.

Point by point response to reviewer's comments:

Reviewer #1 (Remarks to the Author):

Liu and colleagues present data to suggest that endocytosis plays a more important role in regulating TDP-43 levels than autophagy or proteasome function. Their data suggest that modulating endocytosis may be an important therapeutic target for TDP-43 proteinopathies such as ALS and FTD. If true, this would indeed be a novel and significant finding that would change the current model for how TDP-43 levels are regulated. However, their conclusions are based on an artificial overexpression system in yeast, with very limited experiments performed in a *Drosophila* model and in human cells.

Response: *We respectfully disagree with the characterization of the yeast TDP-43 model as “artificial”. It of course is not as physiological as human cell lines, fruit flies or behavior of TDP-43 in ALS patients, hence why we tested our key findings in such systems. However, yeast has proven itself a powerful model for screening for genetic enhancers/suppressors of aggregation-prone proteins implicated in various neurodegenerative diseases. Coupled with cell biology and biochemical techniques, yeast studies have led to new understanding and identification of genetic risk factors in human patients (see introduction). Of course, yeast has also proven time and again to be an excellent model for eukaryotic cell biology.*

Additionally, in comparison to other labs that have published high-impact manuscripts using yeast with a GAL1-driven TDP-43 construct, the degree to which we over-express TDP-43 is less due to our use of lower galactose concentrations (Lines 125-127 + see Fig S1A for physiological consequence of this).

*As to our experiments in human cell lines and *Drosophila* being “very limited”, we note that our original submission featured 19 panels of cell line data, 8 panels of *Drosophila* data and 2 panels of ALS patient data. However, we have added to this further based on the useful comments of all reviewers (9 new panels of cell line data; 2 new panels of fly data) and hope that this will be satisfactory.*

Reviewer opening paragraph continued: Thus, I am far from convinced in their main conclusion that “Endocytosis is a key regulator of TDP-43 toxicity and turnover”, especially since there is significant overlap between the endocytosis and autophagy machinery—indeed Rab5 and UVRAG, the most significant markers to colocalize with and regulate TDP-43 in human cells in their studies, have also been shown to regulate autophagy. Thus, the authors need to show that their means of “enhancing endocytosis” (e.g. overexpressing Rab5) don't also enhance autophagy (which others have observed).

Response: *This is an excellent point. To assess the degree to which Rab5 over-expression enhances autophagy, we examined LC3-I and LC3-II accumulation. Consistent with the results of Ravikumar et al, we observed a small increase in the LC3-II/LC3-I ratio (Fig S7e), though this was far less than what was observed following Rapamycin treatment, which served as a control of robust autophagy induction. As described later in more detail, we also utilized thapsigargin treatment, which blocks autophagy but not endocytosis (Ganley et al, 2011), and observed that in this context, Rab5 over-expression still causes a decrease in TDP-43 levels (Figure 6h). Rab5 OE (in the thapsigargin context) also increased the LC3-II/I ratio slightly, again consistent with the Ravikumar data. In short, our new data indicates that Rab5 may indeed enhance autophagy, albeit modestly in our hands, but that an endocytic effect from Rab5 over-expression is still evident.*

As the reviewer stated, we have made more explicit acknowledgement that endocytosis and autophagic pathways overlap, particularly in mammals (e.g. lines 311-313; line 530-531), and our new model figure (Fig8) reflect this better.

Reviewer opening paragraph continued: Given how unexpected their findings are, the authors also need to do more work in Neurons and in cells expressing endogenous levels of TDP-43 to confirm their findings under more physiologic conditions.

Response: There was already data in our original submission featuring TDP-43 and other proteins being expressed at endogenous levels. Specifically, Figure 6e/Supplementary Fig7c demonstrated that endogenous TDP-43 protein was significantly decreased in abundance when Rab5 or UVRAG was knocked down. Figure 7e and 7f demonstrated endogenous TDP-43 and Rab5 co-localize in ALS patient tissue. Furthermore, Supplementary Figure 7a demonstrated that a HEK293A cell line that we generated to express tagged TDP-43 alleles (WT, Q331K, TDP-35) did so at levels equivalent to endogenous TDP-43 expression. In such cells, defects in cell viability (Figure 6c) were exacerbated with endocytosis inhibition (dynasore), particularly with cells expressing aggregation-prone TDP-43 variants. Conversely, TDP-43-induced defects in viability were suppressed by Rab5 over-expression.

We have now added additional datasets involving TDP-43 at endogenous levels. Specifically, Figures 6g and 6h demonstrate, using chemical and KD approaches, that endogenous TDP-43 is subject to both endocytic and autophagic means of protein turnover (discussed more fully in later comments). Also, although the information was previously in figure legends, we have added more explicit descriptions of when proteins discussed in our results section were expressed endogenously or ectopically to aid the reader (e.g. lines 322-324; 326; 335; 338; 348; 365-366; 370; 387; 393)

However, it is true that much of our cell line data utilizes transfected TDP-43 constructs. However, this is not uncommon in the field for numerous practical reasons, and other highly cited publications that have been key supports for the role for autophagy or the proteasome in TDP-43 clearance also utilized transfection or over-expression TDP-43 models (e.g. Scotter et al, 2014 JCS, Wang et al, 2010, Neuroscience letters). We thus feel our data is comparable to other studies in the field.

As for doing work in Neuron cells (we presume the reviewer means primary neurons; we have limited neuroblastoma data already), we would respectfully suggest that the volume of data we have collected, and now added to, across several model systems is sufficient for the scope of this work. In any case, the specifics of experiments to be conducted in neurons were not clear to us.

In Figure 5j, does Rab5-OE rescue reduced viability induced by TDP-43 expression? This appears to be a critical experiment since Rab5-OE dramatically reduces TDP-43 levels in Figure 5h-i.

Response: This is an excellent suggestion and we did as the reviewer asked – see new data in Fig6c. Consistent with our model, the decrease in cell viability caused by expression of different TDP-43 constructs (at endogenous levels) is significantly rescued by Rab5 over-expression. We describe this new data on lines 352-353.

Also, the authors need to show that Rab5 overexpression doesn't upregulate autophagy given that Ravikumar and colleagues have shown that Rab5 modulates mutant huntingtin toxicity through regulation of autophagy (Ravikumar et al J Cell Sci 2008).

Response: See our response to the second comment of the reviewer above.

In addition, we have added discussion on the parallels between our findings and that of Rubinsztein lab who have linked Rab5 to alleviation of toxicity associated with protein aggregates (lines 499-501 in discussion). We apologize for not including this in our previous submission.

The effects of Rab5 and Rab7 inhibition or overexpression on TDP-43 phenotypes in Drosophila are quite modest. Others have shown that genes that regulate autophagy are also modifiers of TDP-43 in the fly; since the authors claim that autophagy doesn't play an important role in TDP-43, the authors should compare effects of genetic modification of autophagy. Again, since Rab5 overexpression may also enhance autophagy, I am not convinced that their rescue experiments support their conclusion.

Response: Besides a report (Kim SH et al, JBC, 2009) on UBQLN1 expression potentiating TDP-43 toxicity in a fly model (note this gene is also linked to proteasomal function), we are unaware of the data that clearly shows genes that modify autophagy are also modifiers of TDP-43 in the fly. However, we would be grateful if the reviewer can refer us to the appropriate manuscript(s). Regardless, the suggestion to manipulate autophagy in flies and observe the effects on TDP-43-induced motor-neuron dysfunction is a good one. Thus, in this context we have tested the effect of over-expression of Atg8a, which has been previously reported to increase autophagy levels in flies, and enhance clearance of various damaged/ubiquitinated proteins (Simonsen A et al, Finley lab, Autophagy 2008). As shown in Supplementary Figure 8f (lines 422-424), no difference in larval turning times was observed in control, TDP-43 or TDP43-G298S expressing flies +/- Atg8 over-expression. This suggests that the effects we observe with Rab5 and Rab7 in flies are likely to be at least partially independent of autophagy. We also note that although our Rab data may be modest in the reviewer's opinion, they are statistically significant ($p < 0.01$ in most cases).

The limited human ALS data are not convincing, particularly given prior reports that TDP-43 does not localize to endosomes (see Matej et al, Exp Neurol. 2010;225:133-9. "Increased neuronal Rab5 immunoreactive endosomes do not colocalize with TDP-43 in motor neuron disease.")

Response: We are unclear as to what the reviewer finds unconvincing about our finding that 35% of TDP-43 foci co-localize with a Rab5 foci (besides the Matej data, see below). We would welcome a clarification.

Regarding the Matej et al manuscript, although seemingly contradictory, it is hard to make direct comparisons between their work and ours for the following reasons. 1.) The tissue examined is distinct – they look at spinal cord tissue, whereas we examined frontal cortex tissue. 2.) Different Rab5 antibodies were used (Abcam in our study versus Calbiochem in their study) 3.) Matej et al examined phosphorylated TDP-43 co-localization (Cosmo Bio Co antibody) with Rab5 – our TDP-43 antibody (Proteintech) recognizes all forms of TDP-43.

Interestingly though, Matej et al did observe an increase in Rab5 levels in cells from ALS patients, which if our working model is accurate, could reflect an adaptive response of cells either to maintain endocytic activity and/or facilitate TDP-43 clearance. Furthermore, in their electron microscopy datasets, it is not clear if/how the authors define endosomal compartments in their images, though immunogold-labelled TDP-43 does appear on the periphery of vesicular-like structures. We have not determined if TDP-43 association with endosomes is extra or intra-luminal, so our results may not be in conflict.

In addition, the work of Ravikumar indicates that Rab5 and autophagic markers can co-localize, and since several studies (including ours) have shown TDP-43-autophagic marker co-localization (New Fig 5c), one would predict Rab5 should localize with TDP-43 as well if these prior observations are accurate, which we believe they are. Given this contradiction (Ravikumar data v Matej data), the confidence we have in our own data, the differences in experimental approach, and the requirement for speculative guesses as to why differences were seen, we didn't feel discussion of the Matej paper would add positively to the paper. However, if the reviewer believes it is important, we will happily add a brief discussion of the Matej work.

Minor points:

The yeast experiments appear well performed and are convincing. However, more details need to be included on the TDP-43-GFP overexpression model. The authors need to show that the GFP tag doesn't affect the turnover of TDP-43. The authors should provide more details on the construct in methods—importantly, is the human TDP-43 3'UTR present? If so, it may be able to autoregulate its mRNA levels which could confound interpretation of protein stability.

Response: We have added a "plasmids" section to our methods where we refer readers to Supplementary Table S1. Here, we include pertinent information including promoters/5'UTRs, and 3'UTR/terminator sequences.

Clarification: Due to a miscommunication, yeast data labelled TDP-43-GFP in the prior submission should have been written as TDP-43-YFP, as this is in fact what TDP-43 is fused to (plasmid obtained from Johnson et al, Gitler lab, 2009). To be clear, this does not alter data interpretations as GFP and YFP are nearly identical proteins, both are recognized by the GFP antibody we used, and GFP and YFP both behave similarly during vacuolar turnover (i.e. a GFP/YFP fragment persists while the fusion protein partner degrades e.g. Li et al, Vierstra lab, Plant Cell, 2015).

Regarding whether the YFP tag affects the turnover of TDP-43, we obtained a non-YFP tagged TDP-43 construct, and examined its rate of turnover following cycloheximide addition. We saw no difference in the rate of turnover between TDP-43-YFP and TDP-43 (Figure S1b; lines 130-131).

The TDP-43 construct used in yeast does not have the TDP-43 human 3'UTR present. In any case, it is unclear if this UTR would confer the same regulation in yeast as it does in humans, and even if it did affect overall TDP-43 levels by impacting translation/mRNA decay, it would not directly affect protein stability (i.e. a post-translational/mRNA decay measurement). This is what our cycloheximide protein stability data is based on.

Sup fig6 – (a) why is there no colocalization of mutant TDP-43 with transferrin? Colocalization of endogenous Rab5 with transfected TDP-35 and 25 punctae are quite convincing and striking – however, colocalization is much less convincing with overexpression of ALS/FTD-causing TDP-43. At best, they can say that there is “partial” colocalization. In contrast, the colocalization of mutant TDP-43 with endogenous UVRAG is striking. Since UVRAG is implicated in autophagy, might this suggest that these TDP-43 punctae are autophagosomes as has been described in multiple papers? The authors should show autophagosome markers in the same panels—it may be that Rab5 is also mislocalized on autophagosomes in these cells.

Response: *The quality of images selected for this panel were poor, and out of focus – we apologize for this oversight. We have replaced images in Fig S6a, and now highlight areas where TDP-43 foci (Q331K and TDP-35 form foci far more readily than WT) do in fact partially co-localize with transferrin (line 321; Fig S6a).*

Regarding co-localization with the Q331K TDP-43-GFP construct, in general this TDP-43 construct forms fewer, less intense foci compared to the TDP-35 or 25 constructs, thus the co-localizations tend to look less striking. However, we feel the images shown are still convincing for Rab5, Vps34 and UVRAG (S6b, d and e).

UVRAG has been linked to endocytosis as well as autophagy, and based on our reading of the literature, this remains controversial in human cell lines, unlike in yeast for Vps38 (previously commented on line 188). However, to the question of whether TDP-43 foci and Rab5 are localized with autophagosomes, following the reviewer’s suggestion, we have examined and quantified TDP-35 co-localization (as this forms foci more readily than WT and Q331K) with endogenous LC3 and Rab5 foci in the same cells (Fig 5c). Consistent with prior literature, we do see good co-localization of LC3 with TDP-35 foci, (47%, line 323-325). However, the % co-localization of TDP-35 foci with Rab5 is greater (80%). It is indeed the case that some Rab5 and LC3 foci overlap, as we indicate in the figure with an arrowhead. It is possible as the reviewer suggests that Rab5 is “mis-localized” on autophagosomes (though if it functions in autophagosome assembly, this may be a physiological location). It is equally possible based on this data alone that LC3 is “mis-localized” on endosomes. Other possibilities include the functioning of both proteins on endosomes and autophagosomes, some type of hybrid compartment, or a non-functional aggregate that sequester endocytic and, perhaps to a slightly lesser degree, autophagic components. We discuss most of these issues in our discussion (lines 446-448; 507-519; 523-525; 529-534 and have amended our model figure (Figure 8) to reflect these possibilities.

Overall though, since LC3 and Rab5 are commonly accepted markers of autophagosomes and endosomes respectively, and since TDP-43 co-localization is greater with Rab5 than LC3 (in yeast and human cell lines), it seems acceptable to us to suggest that a higher fraction of TDP-43 is associated with the endocytosis pathway than an autophagic pathway. However, to be clear, we also believe that there is some association of TDP-43 with autophagy.

“In contrast, TDP-43-GFP only exhibited a modest decrease in vacuolar turnover and no full-length protein

accumulation (Fig. 1e).” It appears to me in Fig 1e that there is a marked decrease in free GFP, suggesting that there is a reduction in trafficking to the lysosome (or decreased lysosome function).

Response: We have rephrased this sentence as follows (line 165): “In contrast, TDP-43 YFP turnover was not fully reliant on core autophagy genes for vacuolar turnover (Fig. 1e).” We agree that there is a reduction, consistent with either autophagy having a role in trafficking TDP-43 to vacuole, or that vacuole function is impaired, though the physiological significance of this (Fig 1g) seems minimal.

Vps15 E200R effect would be more convincing if they showed Vps15 WT in same panel from same expt.

Response: As requested, we have added Vps15 WT growth assay data to Fig S2b, which was done on the same plate as the original data shown in our previous submission. We simply chose not to show this as we felt it redundant with data in Fig S2a, but are happy to add it back.

How does vps38 overexpression “enhance endocytosis”? Has this increase in endocytosis been shown previously?

Response: This is a good question, and we don’t know the answer. We now rephrase our rationale sentence on lines 233-240 to shift emphasis on testing differential effects of PI3K I and II, and state clearly on lines 246-247 that we do not know why Vps38 over-expression enhances endocytosis rates, though we speculate it may reflect an increase in PI3K II complexes (endocytosis promoting) because Vps38 is rate limiting for complex II formation. Related to this, Vps38 OE could also convert PI3KI to PI3K II complexes (i.e. Vps38 out-competes Atg14 for the other components). To our knowledge, this is the first description that Vps38 over-expression enhances endocytosis in yeast (stated on line 245-246).

In addition, we also demonstrated Vps38’s ability to induce endocytosis is independent of TDP-43 by doing additional FM4-64 uptake assays +/- Vps38 OE (See Fig 3e and S3d; line 243-244). In addition, in addressing a comment of reviewer three on the significance of the effects that we see on yeast endocytosis rates, we have chosen a new quantitation method for our FM4-64 data that focuses on the intensity of vacuolar staining by FM4-64, rather than a binary distinction of whether any vacuolar staining is or is not visible. We feel this better reflects the average behavior of the dye endocytic uptake process, and indeed shows the effects to be slightly more pronounced than our previous quantitation suggested, as one could in fact already perceive if they studied our microscopy images carefully.

In discussion, authors argue that there is “emerging evidence suggesting endocytosis is perturbed in als”. In the genes and the papers that the authors describe, there is significant overlap between the endocytic pathway and autophagy. For example, emerging evidence suggests c9orf72 plays a role in autophagy rather than “endocytosis”.

Response: The reviewer is correct that there is significant overlap between autophagy and endocytosis, and that some of the genes we mentioned have been linked to both pathways. Thus, we have amended our discussion paragraph that originally focused on endocytic perturbation in ALS (now starts line 455); edited to mention endocytic-autophagy links on lines 455-456. We now acknowledge that the Filimonenko paper we cited, and work by Hadano et al, PLOS One, 2010, suggests that ESCRT and ALS2 mutants may cause defects in autophagic clearance due to a failure of autophagosome fusion with the endocytic pathway (lines 461-463) As for C9ORF72, opposite conclusions have recently been reached as to whether depletion of C9ORF72 promotes (e.g. Webster et al, EMBO J, 2016; Sellier et al, EMBO J, 2016) or inhibits autophagic activity (Ugolino et al, PLOS Genetics, 2016), and no clear consensus in either camp seems to have emerged regarding mechanisms, thus we only fleetingly mention this on lines 466-468 given word-limit constraints.

Reviewer #2 (Remarks to the Author):

In the manuscript entitled “Endocytosis is a key regulator of TDP-43 toxicity and turnover” Liu and colleagues show endocytosis as a new clearance pathway for TDP43. Although there is an effort in this study to include several disease models to support the unique role of the endocytic pathway in rescuing TDP43 proteinopathy there are concerns that need to be addressed prior to publication:

1. The authors claim that inhibition of the proteasome alone or together with autophagy has no effect on TDP43-GFP stability. However, there is an extensive literature that demonstrates how TDP43 is cleared through both the proteasome system and autophagy, including chaperone-mediated autophagy (Wang 2010; Huang 2014, Scotter 2014, Barmada 2014).

Response: *We would like to clearly state that we never intended to convey the message that autophagy has no role to play in clearing TDP-43, and in fact our prior submission showed data in yeast and mammals where there were autophagic effects on TDP-43 clearance. We appreciate however that our emphasis on the novelty and strength of the endocytic effect may have made this unclear – to this and all the reviewers, we apologize. We have tried to place greater emphasis throughout the manuscript that autophagy does in fact have some role to play in TDP-43 clearance, both in yeast and human cells. Examples of such edits include:*

line 99 “...and suggests endocytic upregulation as a novel therapeutic strategy of significant potential.” (no mention of greater potential than autophagy modulation).

We have altered result section subheadings on line 309 to acknowledge that TDP-43 localizes with autophagic components.

Acknowledged in results that Rab5 has been linked to autophagic function (line 332-333).

Line 382: “..this data supports prior conclusions that autophagy also aids TDP-43 clearance in human cells.”

Line 388: “Consistent with prior findings, AT5KD indeed caused TDP-43 accumulation..”

Line 394: “..consistent with autophagy facilitating TDP-43 turnover..”

Line 522: “We also see even clearer evidence for a role for autophagy in TDP-43 clearance...”

Line 529: “We hypothesize that TDP-43 aggregate clearance may be enacted by both endocytosis and autophagic pathways, or some type of hybrid mechanism...”

Data in our manuscript that supports a role for autophagy in TDP-43 clearance (indicates new data) includes: Fig 1a-e, Fig S1d,i – yeast; Fig5c*, Fig6f, g*,h* – human cell lines). In short, our data does indicate a role for autophagy in TDP-43 clearance, but a key novelty of our work is that endocytosis also has a role in TDP-43 clearance that it is independent of autophagy.*

1. continued: The authors should resolve this discrepancy by using multiple null strains of the autophagy pathway. Additionally, the authors should inhibit specifically the autophagosome-lysosome fusion with drugs (Eg: Bafilomycin A, ammonium chloride and/or other inhibitors) to further support that autophagy has no effect in clearing TDP43.

We presume from the phrase “null strains” the reviewer is referring to our yeast studies. If so, we have already used multiple autophagy null strains when it comes to our growth assays (Fig 1g = atg1, atg8, S1l = atg7, atg9, atg11, atg18, atg19), where we can at least conclude that blocking autophagy by multiple mechanisms does not have a physiological effect on cell viability in the presence of TDP-43 expression. For our protein stability and turnover assays, we chose to examine just atg1 and atg8 nulls under multiple conditions (see Fig1a, e, Fig S1c, d), as atg1 and atg8 are amongst the most conserved and inhibitory mutants one could choose to block autophagy with.

Our claim that inhibition of the proteasome alone, or together with an autophagy block (*atg1* or *8* null) had no effect on TDP-43-YFP stability was specific to mid-log growth conditions (Fig S1c, d). However, having quantitated our TDP-43 bands, although proteasomal inhibition indeed had no effect, the combination of an autophagy block and proteasomal inhibition did have a minor effect - we have revised line 135 to reflect this.

Chaperone mediated autophagy (CMA) does not occur in yeast to our knowledge, and thus there are not null strains for us to examine this. We do not dispute that CMA likely has a role in human cells in TDP-43 clearance, nor the proteasome for that matter (even if this isn't true in yeast). However, a role for these processes is not mutually exclusive with roles for autophagy and endocytosis. We have made mention of these possibilities in our model figure (Fig 8), but an extensive investigation of these processes in human cells is beyond the scope of this work, and redundant with prior studies.

Regarding inhibiting autophagosome/lysosome fusion with Bafilomycin A or ammonium chloride, as the reviewer used the word lysosome, we presumed they wished us to test the effect of these drugs in mammalian cells. However, the use of the drugs suggested is complicated as numerous papers have indicated that Bafilomycin A1 (and ammonium chloride) inhibits endocytosis, as well as fusion of endosomes and autophagosomes (e.g. see Klionsky et al, *Autophagy*, 2008). However, we attempted this experiment in any case, and although we ultimately omitted it from the paper due to difficulties in interpretation (and our use of Thapsigargin— see below), we describe it here as there is still a minor result consistent with a role for endocytosis.

In our hands, higher doses of Bafilomycin used in some other studies lead to significant cell death in HEK293A cells. In addition, TDP-43 protein is quite a stable protein, thus we ultimately examined the effect of 24 hours of exposure of 0, 5, 10 and 50nM of Bafilomycin A1 to endogenous TDP-43 protein levels; this time allows effects of altered turnover rates to manifest on steady state TDP-43 levels. We also monitored LC3-I and II levels to gauge autophagy activation. As seen in the figure below, TDP-43 protein levels increase at all three concentrations, though at 5nM, there is little/no increase in LC3-II levels, even though TDP-43 levels increase 74%. Although not conclusive alone, this could be consistent with TDP-43 levels increasing due to inhibition of endocytosis in the absence of autophagic inhibition. However, because both autophagy and endocytosis (and CMA) may be perturbed by the drug, we did not include this data in our paper.

Instead of Bafilomycin, we ultimately helped distinguish the contributions of autophagy and endocytosis by using Thapsigargin (Ganley et al, Jiang lab, *Mol Cell* 2011). Ganley et al demonstrated that thapsigargin blocks autophagosome fusion with the endocytic system and lysosomes by blocking Rab7 recruitment to autophagosomes, but leaves endocytosis completely intact. In Fig 6h (lines 391-395), we first show that thapsigargin treatment alone does increase TDP-43 protein levels (and LC3-II), consistent with a role for autophagy in TDP-43 clearance (and an autophagy block - Ganley et al also observed LC3-II accumulation). More importantly, over-expressing Rab5 at the same time reduces TDP-43 protein levels back to control levels, indicating that “a Rab5-enhanced, autophagy-independent mechanism exists that can help clear TDP-43.” (line 396-399). A modest increase in LC3-II is also observed, consistent with Ravikumar et al’s data that Rab5 OE may also increase autophagosome assembly and thus autophagic flux.

In conclusion, using a drug to block autophagy revealed that a.) autophagy does have a role to play in TDP-43 clearance and b.) that an autophagy-independent means of TDP-43 clearance exists that is driven by Rab5 over-expression, which many have shown enhances endocytosis rates (lines 329-331 + refs 43-45). Note, this

data also fits with our observations that even in an Atg5 KD context, which inhibits autophagy, dyansore treatment (endocytosis inhibitor) causes an increase in TDP-43 protein levels (Fig 6g).

2. The total expression levels of TDP43 in the control and TDP43 over-expressing line should be measured and shown.

Response: We are uncertain as to what control and over-expressing lines the reviewer is referring to, or the rationale for this request. We are presuming that the reviewer is referring to the HEK293 lines we generated that express tagged TDP-43, TDP-43 Q331K or TDP-35. We already had data in our previous submission showing that expression of these constructs closely resembled endogenous TDP-43 expression, but we have added an extra lane to show endogenous TDP-43 expression in the absence of tagged TDP-43 genomic integration. As mentioned to reviewer 1, several of our key datasets also utilize endogenous TDP-43 detection.

If we are mistaken in our assumption, we will happily do westerns, and would appreciate specific instructions (e.g. a figure reference).

3. The authors demonstrate that starvation induced-autophagy promotes TDP43 turnover, although the usage of Atg1 and Atg8 mutants do not affect turnover. To the knowledge of this reviewer, non-selective autophagy does involve these Atg proteins. The authors should provide literature to support this data or test multiple null strains and dissect the two autophagy pathways.

Response: Atg1 and Atg8 are indeed essential for non-selective autophagy, and in fact Atg1 and Atg8 mutants do affect TDP-43 starvation induced turnover. The data in Fig 1e (stationary phase) and Fig S1 k (-N stress) demonstrate that atg1 and atg8 deletion partially impair turnover of TDP-43 in vacuoles, as indicated by a partial but not complete decrease in YFP fragment (see comment to reviewer 1). In contrast, we show that vacuolar turnover of Pab1, which we previously identified as an autophagic (granulophagy) target, is completely blocked by Atg1 and Atg8 deletion. This is consistent with Pab1 being solely targeted to vacuoles in an autophagic manner, as we predicted based on our prior 2013 Cell paper.

Thus, we are not asserting that the accelerated turnover of TDP-43 observed in Fig 1C is fully or partially Atg1 or Atg8 independent. Rather, what we are saying is the fact that there is still turnover evident in Fig 1e, even in the absence of Atg1 or Atg8, indicates that there must be a second vacuolar turnover mechanism that is autophagy independent (i.e. endocytosis). We subsequently examine endocytosis null strains for protein levels and stability, and show that in yeast, PI3K mutants (Fig 2D), and several endocytosis genes (Fig 4F) have a more significant effect on TDP-43 levels and stability than atg1 or atg8 mutants.

To be clear, we are not suggesting the existence of two autophagic pathways, rather one canonical autophagic pathway and one endocytic-based mechanism for TDP-43 clearance (which may operate independently, and/or merge to facilitate TDP-43 clearance; see model figure). If we have misunderstood the reviewer's question, we will happily examine other mutants he/she deems relevant.

4. Fig 1f: the increase of TDP43 in the vacuolar mutants is not convincing. The authors should present quantification of the intensity values

Response: This is a good suggestion, and we have quantified the TDP-43 bands, normalized to our Pgk1 loading control. We also repeated our analysis in the prb1 null strain due to uneven lane loading that impacted our previous image – new data for this null is shown in Fig 1f. We hope the data is more convincing and would note that there is likely redundancy in vacuolar protease activity; we have noted this, together with a softening of our language on lines 170-171.

5. Sup Fig 1f: Authors demonstrate that CHX treatment disassembles SG but not TDP43 foci, but the quality of the figure makes the data unconvincing. Images of fluorescence from the single channels and quantification of the intensity values of Pub1 and TDP43 foci might help. Moreover, the intensity of TDP43 foci looks to be

reduced with CHX, which does not support the authors statement that TDP43 foci are resistant, so revisions are necessary.

Response: As mentioned below regarding image quality, we suspect the pdf conversion process may be impacting our images. Nonetheless, we have shown separate channel images as the reviewer requested. We have also quantified the TDP-43, and Pub1 foci for comparison, +/- CHX as the reviewer asked for (S1h). As is clear, Pub1 foci (SG marker) disappear entirely during the CHX treatment, whereas TDP-43 foci are minimally affected. Thus, we believe our original statement is fair.

6. Sup Fig 1h. Authors should show Pab1 induction, so data before and after Dcp2 inactivation need to be included.

Response: We have added a panel of images of the *dcp2-7 atg15Δ Pab1-GFP* expressing strain before Dcp2 inactivation, as requested (Now Fig S1j), which shows minimal accumulation of Pab1-GFP in vacuoles prior to Dcp2 inactivation.

7. Sup Fig 1i. Intensity values of the full-length GFP Pab1 accumulated in Atg1 and Atg8 mutants needs to be shown.

Response: This is a good suggestion and as for the majority of our western blots focused on TDP-43 levels, we have now added quantitation (Now Fig S1k).

8. Fig 4f. Intensity values should be included in bar graphs.

Response: We have determined intensity values. As above, we have shown the intensity values as numbers on the actual western rather than show bar graphs, so that we remain consistent with other figures and save space in our figures.

9. The authors examine the ESCRT complex components and demonstrate that TDP43 toxicity increases in several strains of mutants for different ESCRT complexes. Interestingly the TDP43 has been recently demonstrated to repress the Vps4b, a known ESCRT disassembly factor (Schwenk 2016). The authors should include this protein in their analysis, provide some insight on how TDP43 regulates ESCRT, and include this reference in the discussion.

Response: This is a good suggestion. We have added two pieces of Vps4 data, namely a Vps4-GFP western blot analysis of Vps4 levels in the presence or absence of TDP-43 (Fig S5E), and an analysis of TDP-43 protein stability in a *vps4Δ* mutant (Fig S5F). Like the Schwenk paper which we refer to/reference on line 295, we also see a reduction in Vps4b expression (S5e). We further speculate on how TDP-43 may impact ESCRT or other endocytic function based on the reduction we observe in Vps4b levels (Line 488). We also see that TDP-43 stability is increased in a *vps4Δ* strain (S5f; lines 298-299).

10. The authors show the distribution of TDP43 fragments in Fig 5b. The Rab5 staining in TDP-25 and TDP-35 is not punctate as in the other groups. This looks to be very relevant because of the strong foci reduction observed in figure 5i. Given the importance of TDP43 in recycling endosomes (Schwenk 2016) it is conceivable that these fragments are somehow interfering with the endocytic system. The authors should comment on this observation.

Response: We absolutely agree that TDP-43 aggregates/fragments may interfere with the endocytic system – that is a key conclusion of our paper (e.g. Fig 5a, Fig 3a; discussion sections 484-505). We also observe that expression of aggregation prone TDP-43 fragments (i.e. TDP-35 and TDP-25 – See S6b) particularly inhibits endocytosis rate (Fig 5a). As can be seen by comparing S6b (no Rab5 OE) and 5b (Rab5 OE), there is a significant decrease in the size of TDP-43 foci (quantified in 5d). Our interpretation of this, which we state more explicitly on line 327-329, and 494-504 is that driving endocytosis activity by Rab5 OE may help clear TDP-43

aggregates, which otherwise may sequester and inactivate proteins like Rab5, Hsc70 and/or other endocytic proteins when Rab5 levels are lower. Alternatively, TDP-43 aggregates may alter expression of endocytic proteins given TDP-43's role in gene expression (now stated on 487-490 with Vps4 as an example). At this stage in our study, there are many possibilities, which we hope to explore further in our future studies.

Minor issues

- Improve the image quality of all the figures

Response: We have replaced specific images that were poorly focused (e.g. S6a), and shown new images in select cases to make evidence of co-localization clearer. In the main though, our images were sharp and conformed to resolution guidelines, and we suspect quality may have been lowered by the pdf conversion process.

- Fig 1d. Add scale bar information

Response: Our apologies, the information in the legend for the scale bar was in the wrong place (describing panel b). We have corrected this.

- Sup Fig 1. Add scale bar information

Response: We have done as the reviewer requested.

- Sup Fig 3. Include merged figures in every panel

Response: We have merged our FM4-64 images with our DIC images. We experimented with pseudocoloring our FM4-64 signal to give maximum contrast against our DIC images, but found white to be the best compromise. Note that we have also added data here including a positive control of endocytic inhibition (vps9 null strain, S3a), and evidence of Vps38's ability to induce endocytosis when over-expressed in isolation (S3d)

- Fig 5 b. The Rab5-mRFP and DAPI figures are inverted in the TDP-25 fragment

Response: Thank you for catching this embarrassing error, we have now fixed this.

- Line 299. Use the appropriate language (not man but human)

Response: We have made this change as requested.

- Sup Fig 6b. Rab5 should not have a nuclear staining. The ICC data need to be improved.

Response: We repeated staining for this panel, and now include new images for TDP-35 and TDP-25 expressing lines, as well as including cells with WT TDP-43 (left-most column).

- Sup Fig 6b. Include WT TDP43

Response: We have added this data as requested (see above)

Reviewer #3 (Remarks to the Author):

This interesting manuscript examines the contribution of cellular trafficking pathways to the cytoplasmic aggregation of TDP-43, a nuclear RNA binding proteins associated with ALS pathology. Previous findings indicated a link between autophagy and the clearance of TDP-43 aggregates for detoxification. In this manuscript, a yeast system is used that expresses high levels of TDP-43 to generate toxic aggregates. In their survey of yeast mutants, the authors found that endocytosis mutants had more effect on TDP-43 toxicity than mutants disrupting core autophagy-associated pathways. In particular, associations between TDP-43 and homologs of the endosomal Rab5 GTPase were a focus of this paper, though other endosomal components also seemed to affect TDP-43 toxicity in yeast. Validation experiments were performed in flies and human cells to test whether some of the findings in yeast were generally applicable in metazoans.

These are potentially exciting results here that ascribe a possible mechanism to TDP-43 toxicity and by extension, to ALS pathology. An additional plus to this work was the use of three different cellular/organismal models to validate the functional conservation of the findings. Unfortunately, there were several negatives that outweighed these positives. Weak grammar and lack of thorough explanations led to confusion, especially in the description of this work in the context of previous publications. In terms of experiments, some of the results were overstated, and others experiments required comparative controls and clearer reporting of quantification. Specific comments are provided below.

1. The brief introduction in this manuscript provides little explanation of matters important later in the manuscript. To improve readability, the Introduction could meld disjointed and open-ended sentences into single paragraphs with added description. Throughout the text, details are needed to define central terms and concepts. Readers should be told specifically what Pab1 and Pub1 are, as well as any other key protein that is used as a marker or the relevance of any genes that are being tested. Why is Pab1 a "granulophagy target protein?" More to the point, what generally defines "granulophagy" and what makes it a selective autophagy pathway? How is its mechanism different than other autophagy pathways? To inform general readers, many terms in the Introduction and in later sections need better description.

Response: *We regret that our writing was not clearer. This partially reflects prior editing for submission at other journals, and their word limits, which was restrictive for us given the amount of data we wished to describe. Regardless, we have now provided more detail about Pab1 (line 160-161) and Pub1 (line 148-149), and have defined granulophagy in more detail as the reviewer requested (see below). We have also worked to improve the grammar and explanations in general in the introduction. We are also happy to make/consider any specific edits that the reviewer suggests.*

Overall, we have added additional background in multiple areas including: 1. TDP-43 aggregate toxicity + therapeutic interest in clearing such aggregates (lines 53-61) 2. how SG assembly and mutation of SG proteins may be related to TDP-43 aggregation and toxicity (lines 63-70) 3. Granulophagy, evidence for selectivity, and our initial prediction for its potential effect on TDP-43 aggregate clearance (lines 78-84). The introduction is now approximately 190 words longer as a result. Following the addition of this material, and some alterations in the order in which we presented this information, we hope our new introduction provides a more comprehensive and logical narrative.

2. At first glance, this paper seems to contradict earlier findings the corresponding author published in a previous "Cell" article (Buchan et al. 2013 153, 1461). In that paper "granulophagy" was shown to "clear" stress granules and P bodies, but in this manuscript the analogous TDP-43 aggregates are relatively unaffected by granulophagy/autophagy. The previous paper used yeast functional genomics to identify non-essential genes that clear stress granules, but genes involved in endocytosis genes were not a major group identified. This seems at odds with the results reported here. There would be less confusion if the Introduction section thoroughly explained the previous paper and its findings, and related those results to the issue of TDP-43

aggregation in yeast.

Response: We too were surprised that granulophagy was not responsible for TDP-43 clearance, however we do not believe there is a contradiction as the reviewer perceives. First, in our original work in *Cell* (2013), we were focused solely on stress granule clearance – TDP-43 was not ever expressed in yeast subject to our screen (also contrary to the reviewer's statement, most P-body components were in fact not targeted by granulophagy in yeast – this is also additional evidence of selectivity). Second, our results in this manuscript support the conclusion that yeast stress granule proteins such as Pab1 (and Pub1, Pbp1, eIF4G) are indeed reliant on an autophagic mechanism for turnover during granulophagy inducing conditions (Dcp2-7 inactivation, stationary phase). However, TDP-43, when expressed in yeast, clearly has an autophagy-independent turnover mechanism that leads to vacuolar turnover, and relies on endocytic function. Thus, although TDP-43 can localize in stress granules, turnover of TDP-43 aggregates is at least partially distinct, which we highlighted in our discussion on line 472-474 as an area of intriguing future research. Further consistent with the idea that SG proteins and TDP-43 SG-localized aggregates behave differently, and may not be similarly cleared by cells, it is worth noting that TDP-43 aggregates are cycloheximide resistant, unlike stress granules (Fig S1g; lines 151-154).

As mentioned in the previous comment, we have introduced new details in a paragraph in the introduction section focused primarily on granulophagy, (see line 78-84)

3. In the methods the statement "All images shown are representative of >3 biological replicates" was repeated several times. For the protein blots it seems reasonable for the experiments to have been done in triplicate. However, for the microscopy it is unclear what this statement actually means. Presumably more than 3 cells were examined for determining the percentage of co-localization, though there is no mention of exactly how many cells were counted (n = ?). For quantifying fluorescence intensity ("foci intensity") there was no description in the Methods section for how that was done. Moreover, how many foci were counted from how many cells?

Response: We are now more explicit in what >3 biological replicates means, specifically at least 3 separate experiments in which a minimum of 50 cells per biological sample were examined. Thus, the n is at minimum >150 cells, though in practice it is usually much more than this (See Supplementary materials, line 132-134). For fluorescence intensity, the process we used was described in the methods chapter we referenced – however we have now provided more information about the software used, and process by which intensity of foci is calculated (line 130-132).

Though we are happy to list the minimum amount of cells/foci that we have quantified, to give readers confidence about that robustness of the trends we observe, we have not provided exact information about how many foci were counted from how many cells, or how many cells were counted for each biological experiment. In all papers featuring microscopy data such as this that the PI has published to now, such information has never been requested. In our opinion, we believe, respectfully, that this is because such information is ultimately not substantially more informative than knowing that a sufficient number of cells were examined, which we are confident was the case. To obtain exact cell and foci counts would be laborious, but if the reviewer feels strongly about this, we will oblige.

4. Fig 1f and Lines 129-131: "...TDP-43 stability in *prc1Δ*, *prb1Δ*, *cps1Δ* and *pep4Δ* vacuolar protease mutants during log phase, wherein TDP-43 stability clearly increased..." It is an overstatement to say that stability "clearly" increased for all the mutants. The effect was minor for *cps1Δ* and *prb1Δ* (*prb1Δ* actually seems to enhance TDP-43-GFP degradation at 3 hours), and subtle differences are not definitive without providing quantification of band intensities for each lane relative to the internal loading control.

Response: We agree that quantification of band intensities for each lane relative to the internal loading control is required, and have done this for this figure and other equivalent datasets throughout the paper. Regarding

prb1 nulls, the data previously shown was unevenly loaded, so we now show a better image/alternative replicate for this data (see point 4 reply for reviewer 2).

5. Line 154-156. "Only the *vps38* Δ strain showed enhanced sensitivity to TDP-43 expression (Fig. 2a), suggesting that PI3K complex II was more important in affecting TDP-43 toxicity." Perhaps as shown in Fig 2a it is not a representative result but there is no major difference in growth comparing the *vps38* Δ mutant with the wild-type control. The sensitivities observed with *vps15* Δ and *vps34* Δ are much more robust. Perhaps the combination of *atg14* Δ *vps38* Δ would provide a greater growth defect, especially given the pathway redundancy proposed in Fig 7.

Response: *The reviewer is correct in stating that the *vps38* Δ growth defect is less striking than *vps15* Δ or *vps34* Δ (though we believe the defect is clearly discernible from the *atg14* Δ result and WT). We have now softened our language on line 195-6 regarding the *vps38* Δ result, and as suggested generated a *vps38* Δ *atg14* Δ strain - there appears to be a very minor additive effect (Fig2a). However, *vps15* and *vps34* deletion alone remains the most striking effect. We now briefly comment on this on lines 199-201 indicating that *vps34*, with the aid of *vps15*, may be able to minimally function in promoting endocytosis or autophagy in the absence of either *vps38* or *atg14*. We are confident these results are not due to suppressor mutation artifacts, as has been anecdotally reported for some yeast gene deletion library strains, as we re-engineered all these mutant strains ourselves prior to testing.*

6. Fig 3a, Line 183-184. "...in TDP-43 expressing cells, endocytosis was clearly inhibited..." Because FM4-64 internalization assays can be variable, this experiment should include parallel endocytosis mutant controls. The controls would also provide a comparison to show the degree of inhibition. For these assays, why are stationary-phased cells (i.e. not growing) being used? An independent mean(s) of assaying endocytosis would also be appropriate to verify these results (i.e. lucifer yellow uptake, Ste6 uptake and degradation, etc.), in log-phased cells.

Response: *These are all good suggestions. We repeated our FM4-64 analyses at mid-log, with the presence of a *vps9* Δ strain (*Rab5* GEF), which does not prevent dye entry, but does impede early-late vacuole maturation, and significantly slows vacuolar staining, more so than TDP-43 expressing cells (Fig 3a, Fig S3a; line 224-227). Regarding a second endocytosis assay, we verified our data in Figure 3a with the lucifer yellow uptake assay as the reviewer suggested. We obtained essentially identical results (Fig S3b; line 231-233), in that lucifer yellow dye showed much less uptake to yeast vacuoles in TDP-43 expressing cells versus WT. As with FM4-64, *vps9* Δ exhibited the most striking lucifer yellow uptake defect, consistent with strong *Rab5* inhibition.*

7. Fig 3e, Line 195. "...Vps38 overexpression enhanced the rate of endocytosis in TDP-43 expressing cells..." This is a non-result. There is no real difference, and certainly in stationary phased cells such minor differences would be predicted.

Response: *As requested above we have repeated our analysis in mid-log cells. Furthermore, we realized that our method of quantifying endocytosis rate (namely a binary distinction of are vacuoles with some degree of FM4-64 staining visible or not) did not do justice to rather obvious differences that our microscopy revealed. Thus, we now quantify endocytosis rate based on the intensity of FM4-64 vacuolar membrane staining. This better reflects total endocytic activity in that it measures the amount of the dye that reaches the vacuole, rather than simply whether any dye reached the vacuole. Indeed, the trends we observed previously are now slightly more pronounced than our previous quantitation suggested. The effects of *Vps38* over-expression in a TDP-43 context is also statistically significant. Finally, we additionally examined the effect of *Vps38* OE in isolation, where again we saw a statistically significant increase in endocytic rate (Fig 3e). Thus, we are confident the effects of *Vsp38* on stimulating endocytosis +/- TDP-43 are real.*

8. Fig 4c, Line 233-234. "...three of which (*Vps16*, 18 and 33) increase TDP-43 toxicity when absent (Fig. 4c)..." Contrary to this statement, *vps33* Δ had no effect on TDP-43 toxicity when compared to wild type.

Response: The *vps33Δ* phenotype is weak as the reviewer indicated, and thus we have removed the *vps33Δ* data.

8. Fig. 5c.... Bands are difficult to see and no specific markers for other compartments are shown (Golgi, microsomes, etc.) to prove that TDP-43 doesn't co-fractionate with everything. Fig 5d.... What is the arrowhead pointing at? Fig 5i.... Why aren't TDP-43 foci quantified? No specificity controls for Rab5 overexpression are shown (Rab6 and/or 8?).

Response: The reviewer makes an excellent suggestion regarding us showing a fractionation of multiple compartments, instead of just late and early endosomes. After trialing several methods, we have utilized an Optiprep fractionation protocol from Li and Donowitz *Methods Mol Biol*, (2008) that partially separates ER, Golgi and endosomal fractions, which we identify with antibodies against endogenous Calnexin, GM130 and Rab5 respectively. Note that no single fractionation protocol we have found and tried perfectly separates organelles such as these. We now include this data as a new Fig 5e/Fig S6f (described on lines 333-335), and retain our old data as figure S6g. As can be seen, TDP-43/TDP-35 is localized preferentially with, and most intensely in Rab5 containing fractions, indicating endosomes. TDP-43/TDP-35 shows partial overlap with GM130 fractions, suggesting some potential localization in the Golgi. Finally, a very small amount of TDP-43 is observed in ER containing fractions. Thus, in answer to the reviewers first point, TDP-43 is preferentially localized in endosome-enriched fractions, but can be observed elsewhere too.

Regarding 6a (formerly 5d), the arrow is supposed to be pointing at a TDP-43 cytoplasmic foci that we show as example of the effect of dynasore (TDP-43 foci increase) – we have fixed this error in alignment. Note that we specifically show TDP-43 WT expressing cells here as they have almost no cytoplasmic foci when endocytosis is not impaired (quantified in 6b), thus the relative difference is most obvious here, particularly when zoomed in on a single cell.

Cell line TDP-43 foci are already quantified in various ways. For example, cells with foci = Figure 6b, TDP-43 foci size = 5d; TDP-43 foci co-localization with Rab5/LC3 = 5c inset. If there is other quantiation the reviewer wishes us to add, we will happily do so.

Regarding the specificity controls for Rab5 over-expression, this is a good idea, though we were uncertain as to which datasets the author wished us to test Rab8 over-expression in. We chose to examine Rab8 over-expression effects on TDP-43 protein levels (Fig S7d) and cell viability (Fig S7b), and saw no effect. Although we focused on Rab8 as our specificity control, we also examined Rab6 over-expression and saw no impact on TDP-43 protein levels (we can share this/add to paper if the reviewer feels it important).

9. Fig 6. The Rab5 and 7 mutants examined did appear to affect TDP-43-induced defects in flies. However, without an examination of other endocytosis mutants, other Rab GTPases as specificity controls, and direct measurements of endosomal transport, it is an overstatement to state: "In summary, increased availability of Rab5 and thus presumably endocytic rate can suppress TDP-43 induced phenotypes in an organismal ALS model."

Response: The reviewer makes a good point, therefore we have conducted larval turning assays with LOF Rab8 lines, to maintain consistency with our human cell line data (Line 411-414; Fig S8c). We saw no significant exacerbation of TDP-43-induced larval turning phenotype in a Rab8 LOF context, unlike with Rab5 and Rab7 LOF lines. Time constraints and the loss of personnel from the Zarnescu lab limited us to this additional analysis. Thus, to avoid overstatement, we have also softened our language on lines 402-403 (results header), 418 (conclusion for LOF data) and 428 (the sentence the reviewer quotes). In essence, our edits shift focus to the effects of perturbing/over-expressing endocytic proteins on TDP-43 phenotypes, rather than on the process of endocytosis itself.

A direct measurement of endosomal transport in living flies is not technically feasible to our knowledge. Although we could do this in S2 cells, because we have already measured TDP-43 effects on endocytosis rate in yeast (with 2 assays) and human cell lines, we feel this is not strictly necessary.

Minor Comments:

1. Clarity could be enhanced with better grammar; sentence and paragraph structure could be greatly improved. Long sentences with 4 or more clauses should be separated into several clearer sentences. Paragraphs with 1 or 2 hanging sentences could be amalgamated into proper paragraphs with clear topic and concluding sentences.

Response: *We agree and have tried to ensure our new writing avoids the problems above. Examples of changes that we made (not exhaustive) include:*

Line 48-50: Sentence split into two. Line 95-97: Sentence split into two. Line 123-125: Sentence split into two. Line 164-166: Sentence split into two. Line 195-197: Sentence split into two. Line 208-216: Grouped mini-paragraphs together. Line 314-317: Sentence split into three.

2. There are minor formatting issues with the References list.

Response: *We apologize for this oversight, and have corrected errors to the best of our ability*

REVIEWERS' COMMENTS:

Reviewer #1 (Remarks to the Author):

The authors have satisfactorily addressed my major concerns. I remain skeptical as to their primary conclusion (i.e. that endocytosis is a major regulator of TDP-43 levels), but readers of the paper can draw their own conclusions. Part of my skepticism is the lack of explanation for how TDP-43, a cytoplasmic protein, is taken up by endosomes. The authors note this gap in knowledge in the discussion (lines 471-481)--the authors speculate that TDP-43 may be secreted via exosomes as shown recently (Iguchi et al Brain 2016); unfortunately the authors weren't able to show that TDP-43 was secreted or internalized in their yeast or cell culture model systems to support this model. The other possible model is via MVB formation which seems plausible; however, they argue that TDP-43 colocalizes with Rab5, an early (not late) endosome marker.

Nonetheless this is an interesting paper that has some important findings that will be of interest to those studying TDP-43 and neurodegenerative diseases.

Reviewer #2 (Remarks to the Author):

The finding that TDP43 toxicity can be reduced by enhancing the endocytosis is a conceptual advance in the field although the relationship between autophagy clearance and endocytosis is still unclear in the field. However the points raised by the reviewers have been addressed satisfactorily. The revised version of the manuscript appears to be good and ready for publication.

Reviewer #3 (Remarks to the Author):

I commend the authors for their efforts and hard work in addressing reviewer comments. For the bulk of my comments, reasonable changes were made to address the concerns. For other aspects, I might not fully agree but it is now reasonable to let interested readers decide on the presented data for themselves. One issue, however, still needs attention. A few subjective statements, particularly concerning the microscopy, could be made more robust by simple quantification and analysis of statistical distributions. Although empathetic to the author's sentiment that "To obtain exact cell and foci counts would be laborious...", providing a record of foci and cells counted addresses any worries about overstated results.

Reviewer #1 (Remarks to the Author):

The authors have satisfactorily addressed my major concerns. I remain skeptical as to their primary conclusion (i.e. that endocytosis is a major regulator of TDP-43 levels), but readers of the paper can draw their own conclusions. Part of my skepticism is the lack of explanation for how TDP-43, a cytoplasmic protein, is taken up by endosomes. The authors note this gap in knowledge in the discussion (lines 471-481)--the authors speculate that TDP-43 may be secreted via exosomes as shown recently (Iguchi et al Brain 2016); unfortunately the authors weren't able to show that TDP-43 was secreted or internalized in their yeast or cell culture model systems to support this model. The other possible model is via MVB formation which seems plausible; however, they argue that TDP-43 colocalizes with Rab5, an early (not late) endosome marker.

Nonetheless this is an interesting paper that has some important findings that will be of interest to those studying TDP-43 and neurodegenerative diseases.

Response: *We appreciate the comments and suggestions of reviewer 1 during the review process, and hope to address some of their remaining concerns in future manuscripts. We are pleased that they believe the paper to be "interesting" and to have "important findings" for the field. No additional changes have been made based on these comments.*

Reviewer #2 (Remarks to the Author):

The finding that TDP43 toxicity can be reduced by enhancing the endocytosis is a conceptual advance in the field although the relationship between autophagy clearance and endocytosis is still unclear in the field. However the points raised by the reviewers have been addressed satisfactorily. The revised version of the manuscript appears to be good and ready for publication.

Response: *We appreciate the comments and suggestions of reviewer 2 during the review process. We are pleased that they believe this paper represents a "conceptual advance". No additional changes have been made based on these comments.*

Reviewer #3 (Remarks to the Author):

I commend the authors for their efforts and hard work in addressing reviewer comments. For the bulk of my comments, reasonable changes were made to address the concerns. For other aspects, I might not fully agree but it is now reasonable to let interested readers decide on the presented data for themselves. One issue, however, still needs attention. A few subjective statements, particularly concerning the microscopy, could be made more robust by simple quantification and analysis of statistical distributions. Although empathetic to the author's sentiment that "To obtain exact cell and foci counts would be laborious...", providing a record of foci and cells counted addresses any worries about overstated results.

Response: *We appreciate the comments and suggestions of reviewer 3 during the review process, and their kind words regarding our efforts on the revision.*

As requested, we now provide a new supplemental table (Supplementary Table 2) which lists cell and foci numbers for the various proteins examined in each figure panel. We have made reference to this in the methods section (line 584-585).